# T-box3 is a ciliary protein and regulates stability of the Gli3 transcription factor to control digit number

Uchenna Emechebe[1†], Pavan Kumar P[2†], Julian M Rozenberg[2], Bryn Moore[2], Ashley Firment[2], Tooraj Mirshahi[2], Anne M Moon[1,2,4,3*]

[1]Department of Neurobiology and Anatomy, University of Utah, Salt Lake City, United States; [2]Weis Center for Research, Geisinger Clinic, Danville, United States; [3]Department of Pediatrics, University of Utah, Salt Lake City, United States; [4]Department of Human Genetics, University of Utah, Salt Lake City, United States

**Abstract** Crucial roles for T-box3 in development are evident by severe limb malformations and other birth defects caused by T-box3 mutations in humans. Mechanisms whereby T-box3 regulates limb development are poorly understood. We discovered requirements for T-box at multiple stages of mouse limb development and distinct molecular functions in different tissue compartments. Early loss of T-box3 disrupts limb initiation, causing limb defects that phenocopy Sonic Hedgehog (Shh) mutants. Later ablation of T-box3 in posterior limb mesenchyme causes digit loss. In contrast, loss of anterior T-box3 results in preaxial polydactyly, as seen with dysfunction of primary cilia or Gli3-repressor. Remarkably, T-box3 is present in primary cilia where it colocalizes with Gli3. T-box3 interacts with Kif7 and is required for normal stoichiometry and function of a Kif7/Sufu complex that regulates Gli3 stability and processing. Thus, T-box3 controls digit number upstream of Shh-dependent (posterior mesenchyme) and Shh-independent, cilium-based (anterior mesenchyme) Hedgehog pathway function.

*For correspondence: ammoon@geisinger.edu

†These authors contributed equally to this work

Competing interests: The authors declare that no competing interests exist.

## Introduction

The *T-box* gene family encodes transcription factors that play critical roles during embryonic development, organogenesis, and tissue homeostasis. Mutations in *TBX* genes in humans cause multiple developmental dysmorphic syndromes and disease predispositions (*Naiche et al., 2005*; *Showell et al., 2004*). Heterozygous mutation of *TBX3* causes Ulnar-mammary syndrome (UMS), initially described as a constellation of congenital limb defects, apocrine and mammary gland hypoplasia, and genital abnormalities (*Pallister et al., 1976*). Recently, heart and conduction system defects have also been described in mice and humans with abnormal *Tbx3* (mice) and *TBX3* (humans) function (*Bakker et al., 2008*; *Frank et al., 2012*; *Linden et al., 2009*; *Meneghini et al., 2006*; *Mesbah et al., 2008*). Germline deletion of *Tbx3* in mice results in embryonic lethality with heart, limb, and mammary defects (*Davenport et al., 2003*; *Frank et al., 2012*; *2013*). Tbx3 also regulates pluripotency and cell fate in early development (*Cheng et al., 2012*; *Han et al., 2010*; *Kartikasari et al., 2013*; *Niwa et al., 2009*; *Weidgang et al., 2013*).

TBX3 transcriptional repression controls expression of cell proliferation and senescence factors (*Brummelkamp et al., 2002*; *Kumar et al., 2014a*); abnormal *TBX3* expression occurs in multiple cancers (*Liu et al., 2011*; *Lu et al., 2010*; *Peres and Prince, 2013*). TBX3 also regulates splicing and RNA metabolism (*Kumar et al., 2014b*). Although these studies highlight the important pleiotropic molecular functions of TBX3, little is known about the core pathways it regulates in developing structures that require its function, such as the developing limb.

**eLife digest** Mutations in the gene that encodes a protein called T-box3 cause serious birth defects, including deformities of the hands and feet, via poorly understood mechanisms. Several other proteins are also important for ensuring that limbs develop correctly. These include the Sonic Hedgehog protein, which controls a signaling pathway that determines whether a protein called Gli3 is converted into its "repressor" form. The hair-like structures called primary cilia that sit on the surface of animal cells also contain Gli3, and processes within these structures control the production of the Gli3-repressor.

Emechebe, Kumar et al. have now studied genetically engineered mice in which the production of the T-box3 protein was stopped at different stages of mouse development. This revealed that turning off T-box3 production early in development causes many parts of the limb not to form. This type of defect appears to be the same as that seen in mice that lack the Sonic Hedgehog protein.

If the production of T-box3 is turned off later in mouse development in the rear portion of the developing limb, the limb starts to develop but doesn't develop enough rear toes. When T-box3 production is turned off in the front portion of the developing limbs, mice are born with too many front toes. This latter problem mimics the effects seen in mice that are unable to produce Gli3-repressor or that have defective primary cilia.

Further investigation unexpectedly revealed that T-box3 is found in primary cilia and localizes to the same regions of the cilia as the Gli3-repressor. Furthermore, T-box3 also interacts with a protein complex that controls the stability of Gli3 and processes it into the Gli3-repressor form.

In the future, it will be important to determine how T-box3 controls the stability of Gli3 and whether that process occurs in the primary cilia or in other parts of the cell where T-box3 and Gli3 coexist, such as the nucleus. This could help us understand how T-box3 and Sonic Hedgehog signaling contribute to other aspects of development and to certain types of cancer.

UMS limb phenotypes are variable ranging in severity from hypoplasia of digit 5 to complete absence of forearm and hand (OMIM #181450). Mouse $Tbx3^{tm1Pa/tm1Pa}$ (*Davenport et al., 2003*) and $Tbx3^{\Delta fl/\Delta fl}$ (*Frank et al., 2013*) mutant forelimbs lack posterior digits and the ulna. Hindlimbs of $Tbx3^{tm1Pa/tm1Pa}$ and $Tbx3^{\Delta fl/\Delta fl}$ null mutants have only a single digit, but $Tbx3^{\Delta fl/\Delta fl}$ mutants also have pelvic defects (*Frank et al., 2013*). Embryonic lethality of both types of mutants has prevented elucidation of Tbx3's limb-specific roles.

The Hedgehog pathway is a key regulator of limb development. Shh signaling in posterior mesenchyme promotes digit development and prevents processing of full length Gli3 (Gli3FL) to its repressor form, Gli3R, which constrains digit number (*Litingtung et al., 2002*). The balance of Gli transcriptional activation and repression is critical for proper digit number and patterning (*Cao et al., 2013*; *Hill et al., 2007*; *Litingtung et al., 2002*; *te Welscher et al., 2002*; *Wang et al., 2000*; *2007a*; *Zhulyn et al., 2014*). In mammals, the limited, partial proteolytic processing of Gli3FL to Gli3R requires functional primary cilia, the ciliary protein Kif7 (*Goetz and Anderson, 2010*; *Liu et al., 2005*), as well as balanced activity of Sufu and the ubiquitin ligase adaptors βTrCP and Spop (*Chen et al., 2009*; *Wang and Li, 2006*; *Wang et al., 2010*; *Wen et al., 2010*)

In this study, conditional ablation of *Tbx3* reveals discrete roles for Tbx3 during limb initiation and compartment-specific functions during later limb development to regulate digit number. We discovered a novel molecular function of Tbx3 in the primary cilia where it interacts directly with Kif7 and is in a complex with Gli3. Loss of Tbx3 decreases Kif7-Sufu interactions, resulting in excess Gli3 proteolysis and decreased levels of both Gli3FL and Gli3R. The resulting preaxial polydactyly phenocopies limb defects seen in *Gli3* null heterozygotes and in mutants with abnormal structure or function of the primary cilia (*Cheung et al., 2009*; *Endoh-Yamagami et al., 2009*; *Goetz and Anderson, 2010*; *Haycraft et al., 2005*; *Liem et al., 2009*; *Liu et al., 2005*; *Ocbina et al., 2011*; *Putoux et al., 2011*). Our findings reveal a novel mechanism where Tbx3 in the anterior mesenchyme is required for proper function of the Kif7/Sufu complex that regulates Gli3 stability and processing.

# Results

## Loss of Tbx3 in the limb bud mesenchyme results in preaxial polydactyly and postaxial oligodactyly

*Tbx3* is expressed in discrete anterior and posterior mesenchymal domains in the limb buds from embryonic day (E) 9.5 (**Figure 1A,C,E**, **Figure 1—figure supplement 1**). To assess the role of these

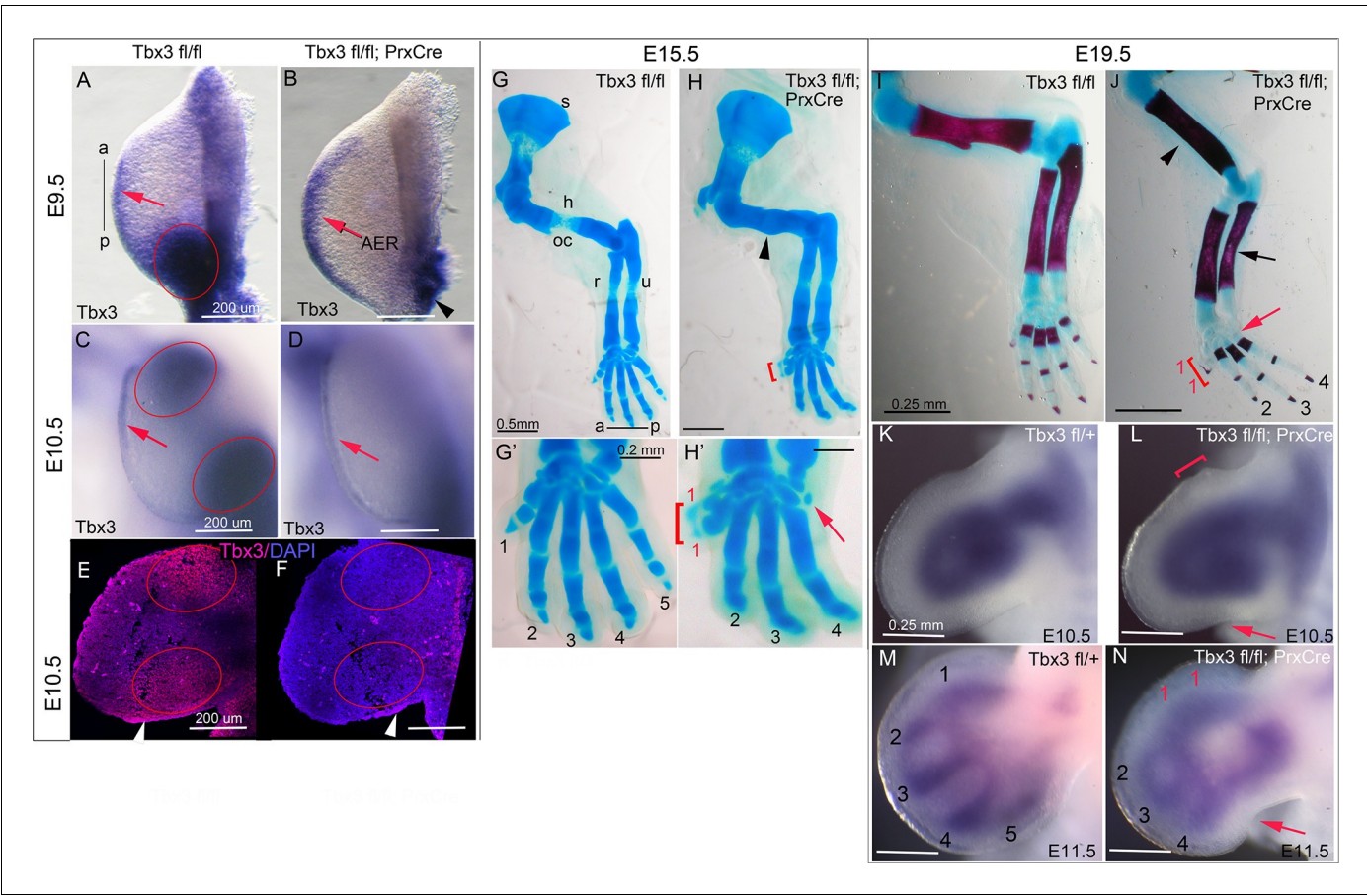

**Figure 1.** Tbx3 regulates anterior and posterior digit development. (**A**) *Tbx3* expression assayed by mRNA in situ hybridization in E9.5 forelimb bud (black line from a-p shows anterior-posterior axis). Red arrow points to *Tbx3* expression in apical ectodermal ridge (AER). Red ellipse encloses posterior mesenchymal expression domain. (**B**) *Tbx3* transcripts are absent in the limb bud mesenchyme of E9.5 *Tbx3^{fl/fl}*;*PrxCre* mutants. *Tbx3* expression persists in the AER (red arrow) and adjacent posterior-lateral body wall (black arrowhead). (**C**, **D**) As in A and B except limb buds are E10.5. Red ellipses enclose anterior and posterior mesenchymal expression domains which are Tbx3 negative in the mutants. Red arrows highlight expression in AER. (**E**, **F**) Tbx3 immunohistochemistry on sectioned E10.5 limb. Tbx3 protein is lost in mesenchyme of *Tbx3^{fl/fl}*;*PrxCre* mutants (F, red ellipses) but AER staining persists as expected (white arrowhead). Please see also **Figure 1—figure supplement 1**. (**G–J**) Skeleton preparations reveal preaxial polysyndactyly (duplicated/fused digit 1, red bracket, H, H', J) and postaxial oligodactyly (absent digit 5, red arrows in H' and J) in *Tbx3^{fl/fl}*;*PrxCre* mutants at E15.5 (H, H') and E19.5 (J). Note delayed ossification of the humerus (H, black arrowhead), loss of deltoid tuberosity (J, black arrowhead) and short, bowed ulna (J, black arrow) in mutant. s, scapula; h, humerus; oc, ossification center; dt, deltoid tuberosity r, radius; u, ulna; digits numbered 1–5 (**K–N**) *Sox9* mRNA expression shows evolving skeletal defects are already evident in *Tbx3^{fl/fl}*;*PrxCre* mutants at E10.5- E11.5. Digit condensations are numbered. Bracket in L shows broadening of digit 1 forming region; red arrows highlight indentation in digit 5 forming region (L, N) and absence of Sox9 digit 5 condensation (**N**).

The following figure supplements are available for figure 1:

**Figure supplement 1.** Ablation of Tbx3 with PrxCre eliminates anterior and posterior mesenchymal protein production.

**Figure supplement 2.** Increased severity of limb phenotypes in *Tbx3* null mutants (*Tbx3^{Δfl/Δfl}*) compared to *Tbx3;PrxCre* is independent of Tbx3 in the AER.

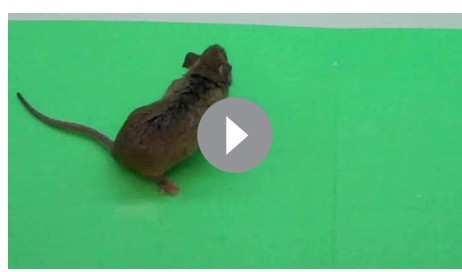

**Video 1.** Adult Tbx3;PrxCre mutant mouse is healthy and mobile despite forelimb deformities.

domains during limb development, we generated conditional mutants using our *Tbx3^flox^* allele (*Frank et al., 2012*; *2013*) and the *Prx1Cre* transgene (*Logan et al., 2002*) (genotype *Tbx3^flox/flox^;Prx1Cre*, henceforth referred to in the text as *Tbx3;PrxCre* mutants). This driver initiates Cre activity at ~14-somite stage (ss) in the forelimb-forming region of the lateral plate mesoderm (LPM) (*Hasson et al., 2007*). Its activity in the hindlimb is irregular, so our analysis focuses on the forelimb. In situ hybridization and immunohistochemistry confirm complete ablation of *Tbx3* mRNA and protein in *Tbx3;PrxCre* mutant forelimb mesenchyme by E9.5 (*Figure 1B–F*, *Figure 1—figure supplement 1*). Expression in the apical ectodermal ridge (AER) is preserved (*Figure 1B,D,F*). We previously reported the specificity of the custom anti-Tbx3 antibody used here and loss of limb mesenchymal protein production in *Tbx3;PrxCre* mutant forelimbs (*Frank et al., 2012*; *2013*).

Unlike mid-gestational lethality seen in constitutive *Tbx3* mutants (*Davenport et al., 2003*; *Frank et al., 2013*), *Tbx3;PrxCre* mutants survive to adulthood with forelimb defects (*Video 1*): 100% have bilateral preaxial polysyndactyly of digit 1 (called PPD1 in humans [*Materna-Kiryluk et al., 2013*]), and 70% lack digit 5 (*Figure 1G–J*). Loss of digit 5 was bilateral in 6/18 and in the remaining, only affected the left forelimb (*Table 1*). This is not due to asymmetric activity of PrxCre because it is also observed in *Tbx3^Δfl/Δfl^* mutants (^Δfl^ = recombined floxed conditional allele) where no Cre activity is involved (*Frank et al., 2013*). Delayed ossification of the humerus (*Figure 1H*) and loss of the deltoid tuberosity (*Figure 1J*) were also observed. Abnormal limb bud morphology is evident by E10.5 (*Figure 1L*) and evolving skeletal defects at E11.5 by the altered pattern of *Sox9* expression (*Figure 1N*).

## *Tbx3;PrxCre* mutant limb defects are less severe than constitutive nulls

Germline *Tbx3* null mutants (genotype *Tbx3^Δfl/Δfl^*) have more severe forelimb defects than *Tbx3;PrxCre* conditional mutants: of the few *Tbx3^Δfl/Δfl^* mutants that survive to E13.5, 100% have agenesis of the ulna and digits 3–5 (*Figure 1—figure supplement 2B*). Their hindlimbs have a single digit and no fibula (*Frank et al., 2013*), phenocopying *Shh* and *Hand2* null mutants (*Galli et al., 2010*). Variable timing of *Tbx3* loss of function by *PrxCre* may account for the disparate forelimb phenotypes of *Tbx3^Δfl/Δfl^* and *Tbx3;PrxCre* mutants, however, our skeletal data and phenotypes of *Tbx3^Δfl/fl^;PrxCre* mutants indicate that such variability manifests as incomplete penetrance of the ulnar and digit 5 defects (*Figure 1 H, H',J*, and Colesanto et al., in preparation).

The AER is a critical signaling center, and *Tbx3* expression is preserved in the AER of *Tbx3;PrxCre* mutants (*Figure 1B,D,F*; *Figure 1—figure supplement 1*, *Frank et al., 2013*). We tested whether AER Tbx3 has a required function using two Cre drivers: *RarbCre* (active in AER and mesenchyme from E9.0 [*Moon and Capecchi, 2000*]), and a novel *Fgf8^mcm^* allele, which produces tamoxifen-

**Table 1.** Increased severity of left limb defects in Tbx3;PrxCre mutants.

**Skeletal phenotypes: E13.5-adult**

| Loss of digit 5 | Bilateral | Left only | Right only |
|---|---|---|---|
| *Tbx3* ^fl/+ or fl/fl^ | 0 | 0 | 0 |
| *Tbx3^fl/fl^;PrxCre* | 6 | 12 | 0 |

**Molecular phenotypes: gene expression**

| Expression pattern or level | Left = Right | Left >Right | Right > Left |
|---|---|---|---|
| *Tbx3* ^fl/+ or f/fl^ | control | control | control |
| *Tbx3^fl/fl^;PrxCre* | 19 | 30 | 1 |

inducible Cre in *Fgf8* expression domains (Moon et al., in preparation). Tamoxifen induction at E8.5 induces robust Cre activity in the AER in RosaLacZ/+;*Fgf8*$^{mcm/+}$ embryos (*Figure 1—figure supplement 2E,F*) and ablates *Tbx3* from forelimb AER by at least E9.5 (*Figure 1—figure supplement 2G–J*). *Tbx3*$^{fl/fl}$;*Fgf8*$^{mcm/mcm}$ mutants have normal limbs (*Figure 1—figure supplement 2L*) and phenotypes of *Tbx3*$^{fl/fl}$;*PrxCre* and *Tbx3*$^{fl/fl}$;*RarbCre* are indistinguishable (*Figure 1—figure supplement 2D* versus N). The results with both Cre drivers indicate that the severe phenotypes of *Tbx3*$^{\Delta fl/\Delta fl}$ mutants are not due to a required function of *Tbx3* in the AER.

## Tbx3 is required for normal limb bud initiation and *Tbx5* expression in the LPM

We next tested whether discrepant forelimb phenotypes in *Tbx3*$^{\Delta fl/\Delta fl}$ and *Tbx3;PrxCre* mutants reflect a role for *Tbx3* in an earlier expression domain than affected by *RarbCre* or *PrxCre*. Limb initiation in the LPM requires *Tbx5* expression in the prospective forelimb territory as early as the 8ss (*Minguillon et al., 2005*), upstream of *Hand2* (*Agarwal et al., 2003*). *Tbx3* is expressed in the LPM from E7.5 (*Figure 2A–C'*). Lineage tracing with a novel *Tbx3*$^{mcm}$ allele (Thomas et al., in preparation) revealed that *Tbx3*-expressing progenitors in the LPM at E8-8.5 give rise to most E10 forelimb mesenchyme in *Tbx3*$^{mcm/+}$;*Rosa*$^{LacZ/+}$ embryos (*Figure 2D*). Consistent with a role for Tbx3 in limb initiation, *Tbx3*$^{\Delta fl/\Delta fl}$ mutants have decreased LPM expression of *Tbx5* (*Figure 2E,F*), visible defects in forelimb initiation and early limb bud morphology (*Figure 2G,H*), and disrupted expression of *Hand2* (*Figure 2I,J*). In contrast, early stage *Tbx5* expression and limb bud initiation are unaffected in *Tbx3;PrxCre* mutants (*Figure 2—figure supplement 1B,B'*)

## TBX3 positively regulates posterior digit development via the Hand2/Shh pathway in posterior mesenchyme

Our data reveal required functions for Tbx3 in limb initiation (demonstrated by *Tbx3*$^{\Delta fl/\Delta fl}$ mutants) and in later limb bud morphogenesis (demonstrated by *Tbx3;PrxCre* mutants). Most *Tbx3;PrxCre* mutants lack digit 5, whose specification and formation depend on 'early phase' Shh signaling beginning at E9.5 (*Harfe et al., 2004*; *Scherz et al., 2007*; *Zhu and Mackem, 2011*; *Zhu et al., 2008*). Hand2 protein is required to activate *Shh* expression in the limb bud (*Benazet and Zeller, 2009*; *Galli et al., 2010*). We found that *Hand2* transcripts are reduced in E9.5 and E10.5 *Tbx3;PrxCre* mutant forelimb buds (*Figure 2—figure supplement 1D,D'*; *Figure 3A, A',F*) as is *Shh* expression (*Figure 3B,B',F*; *Figure 3—figure supplement 1*). Expression of two targets and effectors of Shh signaling, *Ptch1* and *Grem1*, is also markedly reduced (*Figure 3C',E'*; *Figure 3—figure supplement 2*). *Tbx3* expression in posterior limb mesoderm begins earlier than in the anterior compartment (*Figure 1A,C*) and is required for normal *Hand2* in posterior mesoderm (*Figure 3A*, *Figure 2J* and *Figure 2—figure supplement 1D'*) (*Rallis et al., 2005*). Thus, intact *Tbx5* expression in *Tbx3;PrxCre* E9.5 forelimbs (*Figure 2—figure supplement 1B'*) indicates that post-initiation, Tbx3 functions downstream of Tbx5 and upstream of *Hand2*.

To obtain a more comprehensive view of the transcriptional consequences of loss of Tbx3 in limb bud mesenchyme, we assayed gene expression of E10.25 limb buds (32–34 somite stage) by microarray. *Shh* and other hedgehog pathway members and target genes (*Lettice et al., 2003*; *Probst et al., 2011*; *Vokes et al., 2008*; *McGlinn et al., 2005*)(*Lewandowski et al., 2015*) were present among the significantly dysregulated transcripts (*Supplementary file 1*) consistent with the previous report of decreased *Shh* expression in *Tbx3*$^{tm1Pa/tm1Pa}$ mutant limb buds (*Davenport et al., 2003*). In addition to decreased levels of Shh- activated transcripts (*Gli1, Hand2, Osr1, Dkk1, Tbx2, Cntfr, Pkdcc*), we noted increased *Rprm, Zic3, Hand1,* and *Gli3* transcripts and confirmed these with qPCR and in situ hybridization at E10.5–11 (*Figure 3*, *Figure 3—figure supplement 4*). The increase in expression of these putative targets of the Gli3 repressor (*Lettice et al., 2003*; *Probst et al., 2011*; *Vokes et al., 2008*; *McGlinn et al., 2005*) was intriguing as it suggested the possibility of alterations in both Gli3 activator and repressor function in *Tbx3;PrxCre* mutant forelimbs.

The transcriptional profiles of anterior and posterior limb mesenchyme are quite different: alterations in gene expression in either compartment in response to Tbx3 could mask some changes in the other if assayed simultaneously. Thus, we proceeded to microdissect anterior and posterior limb segments and assayed them independently using RNA-sequencing on 38–42 somite stage (~E11;

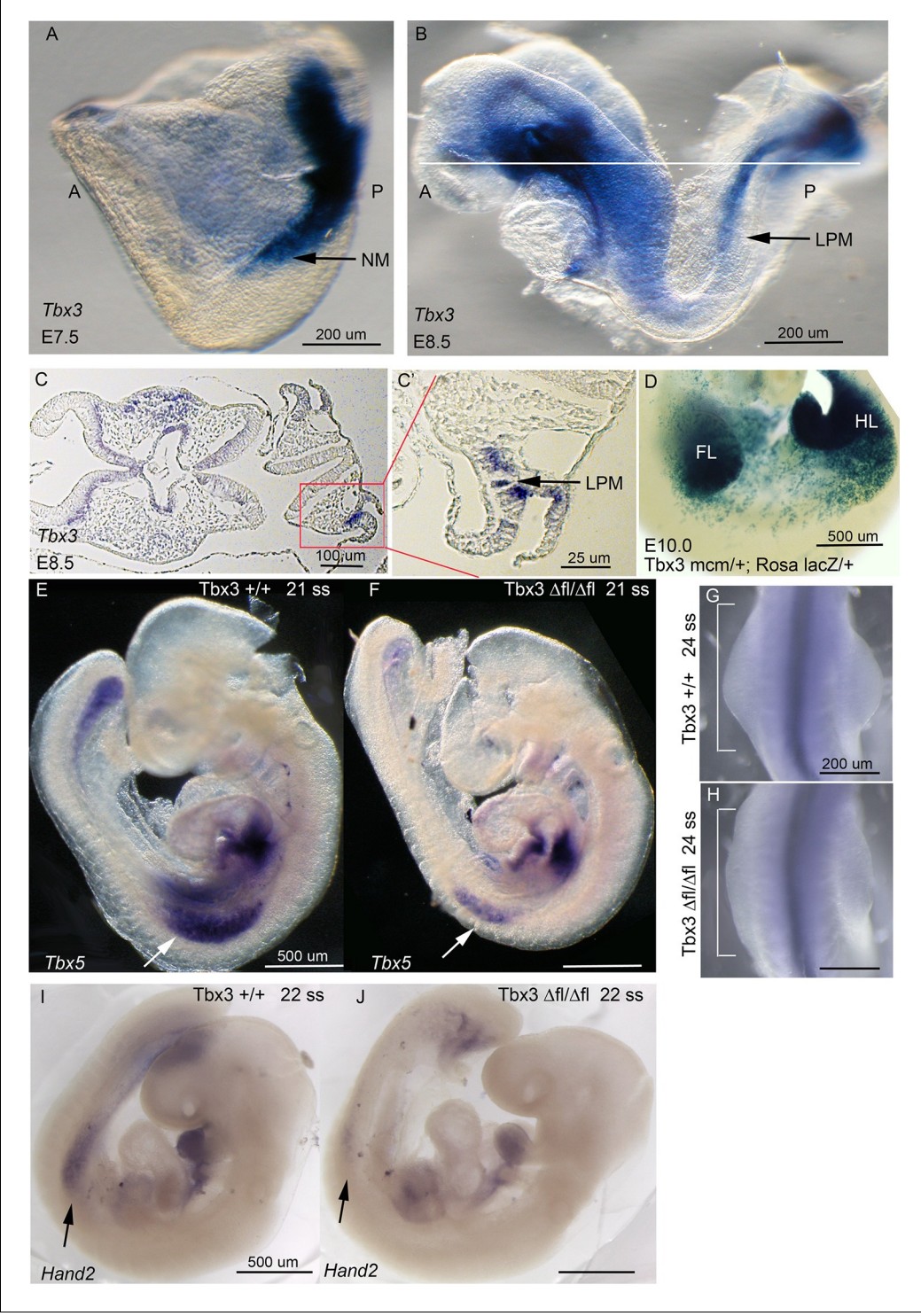

**Figure 2.** Tbx3 is required for normal limb bud initiation. (**A**, **B**) *Tbx3* expression at E7.5 (**A**) and E8.5 (**B**). Anterior on left (**A**), posterior on right (**P**). NM (black arrow) indicates nascent mesoderm exiting primitive streak in panel **A**. (**C**, **C'**) *Tbx3* expression in the LPM (black arrow) of sectioned E8.5 embryo. Plane of section indicated by line in **B**. Panel D is magnification of red-boxed area in **C**. (**D**) X-gal stained E10.0 *Tbx3*<sup>MCM/+</sup>; *Rosa*<sup>LacZ/+</sup> embryo after Cre induction at E8.5. FL, forelimb; HL, hindlimb. (**E**, **F**) 21 somite stage (ss) embryos assayed for *Tbx5* mRNA. White arrows denote forelimb bud. Left sided view. (**G**, **H**) Dorsal view of budding forelimbs of 24 ss embryo forelimbs (neural tube stained for *Shh* expression). Note abnormal shape and size of *Tbx3*<sup>Δfl/Δfl</sup> mutant forelimb buds

*Figure 2 continued on next page*

Emechebe *et al*. eLife 2016;5:e07897. DOI: 10.7554/eLife.07897

*Figure 2 continued*
indicative of disrupted initiation; white brackets are of equal size in both panels. (I, J) 22 ss embryos assayed for *Hand2* expression. Black arrows denote emerging forelimb bud.
The following figure supplement is available for figure 2:

**Figure supplement 1.** Early *Tbx5* expression is normal in *Tbx3^{fl/fl};PrxCre* mutants.

this later stage was needed in order to obtain sufficient RNA from accurately microdissected limb segments) wild type and *Tbx3;PrxCre* mutant forelimb buds (*Figure 3—figure supplement 3*). The resulting RNA-Seq data confirmed accurate dissection with the expected distribution of known compartment-specific transcripts (*Figure 3—figure supplement 3* and *Supplementary file 2*). *Shh*, *Fgf4* and anterior *Hoxd* family transcripts were over-represented in the posterior compartment, and *Alx* and *Pax* family members in the anterior.

Largely consistent with the previous microarray findings, we found evidence of aberrant cell differentiation/fate of posterior mesenchyme with downregulation of Shh-activated targets (*Osr1, Dkk1, Tbx2, Cntfr, Ptch2*, *Supplementary file 3*) that validated by qPCR (*Figure 3—figure supplement 4*). It is known that *Gli3* expression increases with decreased Shh activity (*Wang et al., 2000*), as we see here (*Figure 3D', F*). Although decreased levels of *Hand2, Shh, Ptch1 and Grem1* are clearly evident by in situ and qPCR at this stage (*Figure 3* and *figure upplements*), they were not detected on the RNA-Seq analysis for unclear reasons.

Shh signaling is required for proliferation to ensure sufficient cell numbers to form the normal complement of digits, and loss of Shh results in an increase in the number of cells in G1 arrest (*Zhu et al., 2008*). Assay of cell proliferation in E10.5 and E11.5 limb buds using anti-phosphohistone H3 immunohistochemistry revealed that at E10.5 there was a statistically significant decrease in the fraction of proliferating cells in the posterior mesenchyme (*Figure 3K–M*). At E11.5, proliferation was significantly decreased in both the anterior and posterior mesenchyme, indicating a global reduction in the number of mitotic cells in mutants (*Figure 3N–P*). This suggests that 5th digit agenesis is attributable, at least in part, to decreased cell number, as opposed to decreased proliferation specifically in digit 5 progenitors. Assay for apoptosis using TUNEL showed normal levels of anterior AER cell death at E10.5, as we have previously reported in this region (*Moon and Capecchi, 2000*) (*Figure 3K,L*).

Proliferation of limb mesenchyme depends on activity of FGF8 and FGF4 from the AER (*Boulet et al., 2004*; *Moon and Capecchi, 2000*; *Sun et al., 2002*) and integrity of this structure requires Shh activity in posterior mesenchyme (*Chiang et al., 2001*). Despite decreased *Shh* expression in *Tbx3;PrxCre* mutants, *Fgf4* transcripts were increased in posterior mesenchyme while decreased in anterior (detected by RNA-Seq, *Supplementary files 3,4* and qPCR, *Figure 3—figure supplement 4*), the latter consistent with an expanded digit 1 region. qPCR detected increased *Fgf8* expression in the posterior AER (*Figure 3—figure supplement 4*). Despite these changes in transcript levels, there was no evidence of altered downstream FGF signaling as expression of *Etv4, Etv5, Dusp6* and *Sprys* was unchanged (*Figure 3—figure supplement 5*, note these transcripts are not listed in *Supplementary files 3 or 4* because they did not meet criteria for differential expression). We conclude that despite the decrement in Shh pathway activity in posterior mesenchyme of *Tbx3;PrxCre* mutants, the level is sufficient to maintain ectodermal FGF signaling, consistent with preserved limb outgrowth.

## Tbx3 stabilizes Gli3 protein in anterior limb mesenchyme

To understand the cause of the anterior PPD phenotype in *Tbx3;PrxCre* mutants, we pursued molecular mechanisms known to cause this defect in humans and mice: ectopic Hedgehog pathway activity (*Hill et al., 2007*; *2003*; *Lettice et al., 2003*); decreased Gli3R activity (*Hill et al., 2009*; *Naruse et al., 2010*; *Wang et al., 2007a*); and abnormal composition or function of the primary cilia (*Goetz and Anderson, 2010*).

RNA-Seq analysis of control versus mutant anterior compartments (*Supplementary file 4*) showed no evidence of ectopic hedgehog activity in *Tbx3;PrxCre* mutants, and this was confirmed by in situ hybridization and qPCR for *Shh* and *Ptch1* (*Figure 3B–C'*, *Figure 3—figure supplements 1* and

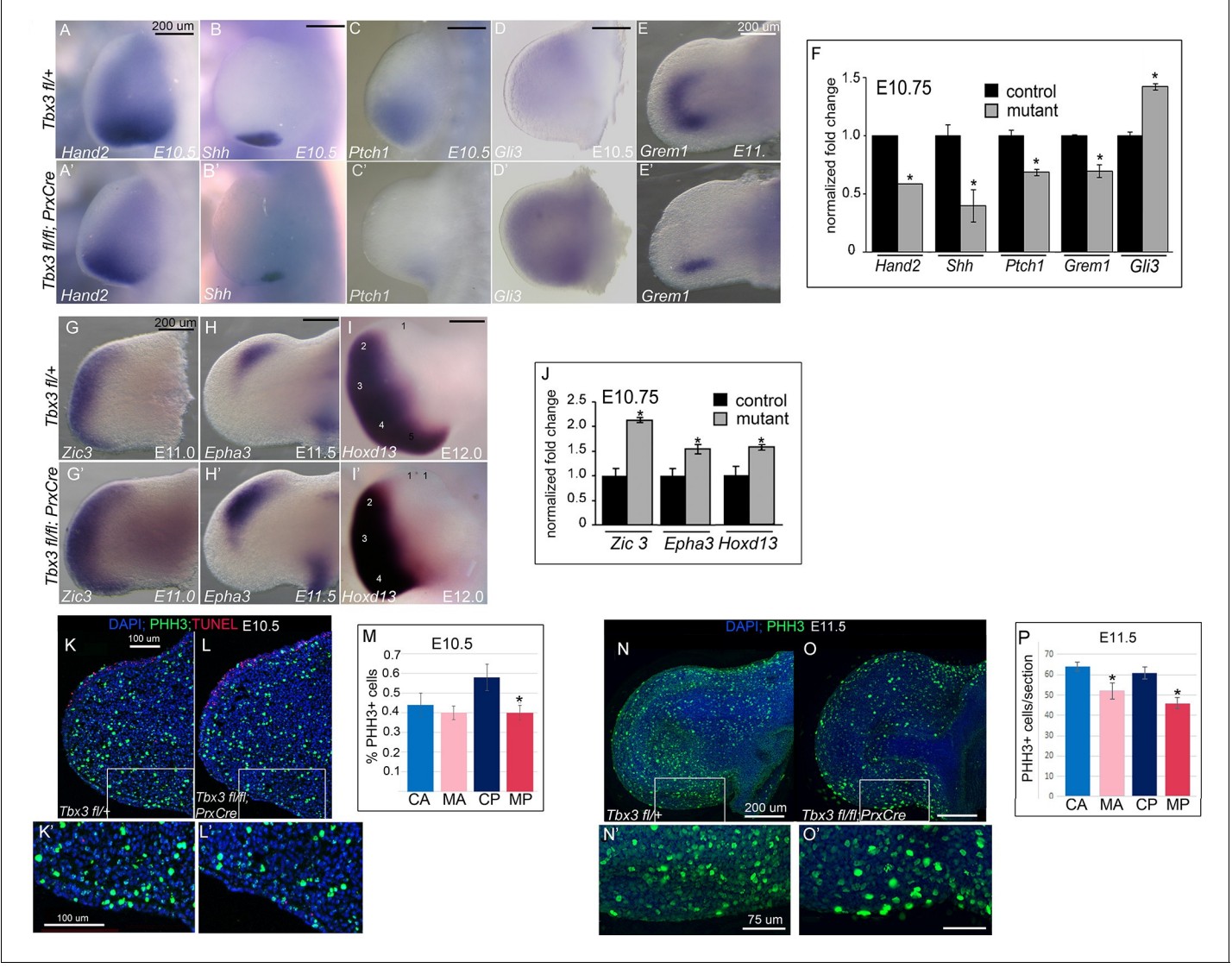

**Figure 3.** Loss of mesenchymal Tbx3 disrupts Shh signaling in the posterior limb bud and decreases Gli3 protein stability. (A–E') In situ hybridization of control and mutant forelimb buds with probes and at embryonic stages as labeled. (F) qPCR of E10.75 (36-39ss) limb buds for transcripts listed confirms findings by detected by in situ. (G–I') In situ hybridization for *Zic3, Epha3* and *Hoxd13* transcripts in forelimb buds of *Tbx3* [fl/+] controls (K–M) and *Tbx3; PrxCre* mutants (G'–I') at ages noted on panels. J) qPCR assay of *Zic3, Epha3, Hoxd13* transcript levels confirms findings detected by in situ. (K–L') Representative images of E10.5 forelimb buds stained for DAPI (blue), pHH3 (green), TUNEL (red). K is *Tbx3* [fl/+] control and K' is digital zoom of posterior mesenchymal boxed area in K. Panel L is *Tbx3;PrxCre* mutant and L' is digital zoom of boxed area in L. This experiment is representative of data obtained from five biologic replicates. (M) Quantification of proliferating cells in anterior and posterior mesenchymal regions encompassing digit 1 and digit 5 progenitors from 20 control and 15 mutant sections. *p=0.02. Control anterior limb (CA), *Tbx3;PrxCre* mutant anterior limb (MA), control posterior (CP), and mutant posterior (MP). (N–O') Representative images of E11.5 whole mount forelimb buds stained for DAPI (blue) and pHH3 (green). N is *Tbx3* [fl/+] control and N' is digital zoom of boxed area. Panel O is *Tbx3;PrxCre* mutant and O' is digital zoom of boxed area in O. Note decreased pHH3+ cells in mutants, particularly cells in prophase and anaphase, which have the faint and speckled patterns compared to the bright staining of highly condensed S-phase chromatin. (P) Quantification of proliferating cells in anterior and posterior mesenchymal regions encompassing digit 1 and digit 5 progenitors from 50 control and 44 mutant sections at E11.5. *p<0.1. Control anterior limb (CA), *Tbx3;PrxCre* mutant anterior limb (MA), control posterior (CP), and mutant posterior (MP).

The following figure supplements are available for figure 3:

**Figure supplement 1.** Decreased Shh expression in E10.5 forelimb buds of Tbx3;PrxCre mutants.

**Figure supplement 2.** No evidence of ectopic hedgehog pathway activity in Tbx3;PrxCre mutant forelimbs.

*Figure 3 continued on next page*

*Figure 3 continued*

**Figure supplement 3.** Microdissection of E11 forelimb buds into anterior and posterior compartments for gene and protein expression analyses.

**Figure supplement 4.** qPCR of additional key transcripts in anterior and posterior forelimb compartments at E10.5–10.75 (36–39 somite stages).

**Figure supplement 5.** *Fgf8* expression and downstream in *Tbx3;PrxCre* mutant forelimb buds.

*2*). Although *Gli3* transcripts were increased in mutant limb buds (*Figure 3D',F*; *Supplementary file 1*), targets of Gli3R transcriptional repression such as *Zic3, Epha3, Hoxd13* (*McGlinn et al., 2005*; *Vokes et al., 2008*) were overexpressed when assayed by microarray, RNA-Seq, qPCR and in situ hybridization (*Figure 3 G-J*, *Supplementary files 1,3,4*).

The discrepancy between *Gli3* RNA levels and increased expression of some repressor targets prompted examination of Gli3 protein levels. Gli3R constitutes the vast majority of Gli3 protein species in the anterior limb bud (*Wang et al., 2000*) and *Figure 4A*). Gli3R protein was markedly decreased (7.4 fold on this representative immunoblot) with multiple bands of lower molecular weight than Gli3R present specifically in *Tbx3;PrxCre* mutant anterior mesenchyme (mutant anterior compartment: MA, *Figure 4A*, red box). Gli3FL was virtually undetectable in mutant anterior mesenchyme (*Figure 4A'*). This finding is not due to poor sample quality because it was reproducible (N=3), no degradation was present in simultaneously prepared posterior compartment lysates (mutant posterior, MP), and the β−tubulin control was intact. These findings indicate that Tbx3 is required for stability of Gli3FL and Gli3R proteins in the anterior limb mesenchyme, and are consistent with the PPD phenotype observed here and in other models of Gli3R deficiency (*Hill et al., 2009*; *Naruse et al., 2010*; *Wang et al., 2007a*).

## TBX3 interacts with Kif7 and is required for normal Kif7 trafficking

In an independent experiment to identify Tbx3 interacting partners, we performed Tbx3 co-immuno-precipitation (co-IP) on E10.5 mouse embryo lysates, followed by mass spectrometry (*Kumar et al., 2014b*). Surprisingly, Kif7 was among the Tbx3 interacting proteins identified. Kif7 is a ciliary protein that modulates activity of the Hedgehog pathway (*Hui and Angers, 2011*). It is required for proper formation of a 'cilium tip' compartment that regulates Gli function (*He et al., 2014*), and for the regulated, partial proteolytic processing of GliFL to Gli3R (*Chen et al., 2009*; *Cheung et al., 2009*; *Endoh-Yamagami et al., 2009*; *Law et al., 2012*; *Ryan and Chiang, 2012*). As in *Tbx3;PrxCre* and *Gli3+/-* mutants (*Hill et al., 2009*), human and mouse *Kif7* mutants have PPD (*Cheung et al. 2009*; *Endoh-Yamagami et al. 2009*; *Putoux et al. 2011*).

Immunoprecipitation of protein lysates from E10.5 forelimb buds showed that Tbx3 co-immuno-precipitates (co-IPs) with Kif7, confirming interaction of Kif7 and Tbx3 in the developing limb (*Figure 4B*; specificity and efficiency of Kif7 IP in this experiment is shown in *Figure 4C*). We next tested whether these proteins directly interact by overexpressing Flag-tagged Kif7 and Myc-tagged Tbx3 in HEK293 cells and immunoprecipitating for either Flag or Myc, followed by immunoblotting for Tbx3 (*Figure 4D*) or Kif7 (*Figure 4E*). Both experiments confirmed direct interaction of the tagged proteins.

The interaction of Tbx3 with Kif7, and the shared PPD phenotypes of *Kif7* and *Tbx3;PrxCre* mutants, suggest that Tbx3 may be required for normal Kif7 function in the anterior limb bud, and that loss of Tbx3 may disrupt Gli3 stability and processing in part via a Kif7-dependent mechanism.

We examined Kif7 localization in E11 control and mutant forelimb buds, co-staining for the cilia marker Arl13b. Confocal fields spanning the anterior mesenchyme of controls and mutants were imaged and *Figure 4F and G* are representative 40X fields (a higher magnification image and confocal z-stack, which also show Kif7 in the cytoplasm and nucleus are shown in *Figure 4—figure supplement 1* and *Figure 4—source data 1*). Blinded to genotype, fields were scored for ciliary Kif7 immunoreactivity as a single puncta, multiple punctae/streak, or none (N=1785 and 1792 cilia scored in controls and mutants, respectively). In 16% of control cilia, Kif7 was detected in two punctae (presumed base and tip) or as a streak along the cilia, but this was only the case in 6% of mutant cilia (*Figure 4H*, p<0.001). There was no significant difference in the number of Kif7+ cilia rather, there were more single puncta cilia in mutants than controls (*Figure 4H*, 84% vs 71%, p<0.001). We did

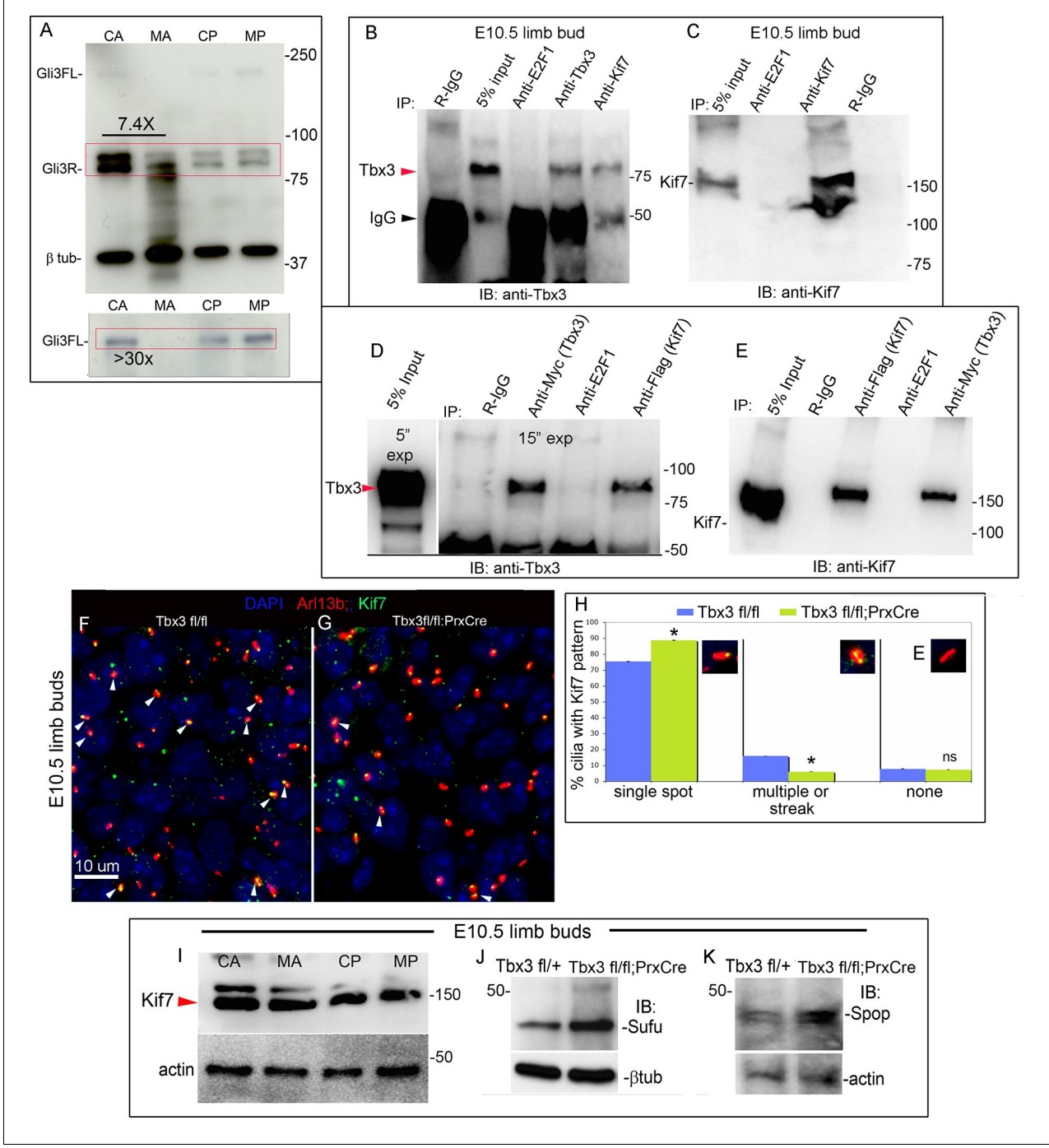

**Figure 4.** Loss of Tbx3 results in Gli3 protein instability and aberrant localization of Kif7 in limb bud cilia. (**A**) Representative immunoblot (N=3) blot of E10.75 forelimb bud lysates prepared from microdissected *Tbx3fl/+* control anterior limb (CA), *Tbx3;PrxCre* mutant anterior limb (MA), control posterior (CP), and mutant posterior (MP) probed for Gli3 and βtubulin loading control. Note decreased level of Gli3FL and Gli3R, and multiple bands of lower molecular weight than Gli3R in MA sample. Densitometry of Gli3R bands in red box in N revealed that in this representative experiment, the level of Gli3R was 7.4 fold decreased in mutant anterior relative to control anterior. (**A'**) Longer exposure of top of blot shown in panel A to examine Gli3FL band. The control (CA) Gli3FL band is 31 fold more intense than mutant (MA, virtually undetectable). (**B**) Immunoblot of lysates from E10.5 forelimb buds immunoprecipitated (IB) with antibodies listed at top and immunoblotted (IB) for Tbx3. Lane 5 shows that immunoprecipitation with anti-Kif7 antibody co-IPs Tbx3. (**C**) As in panel B, but assayed for Kif7. (**D**) Co-IP assay of Myc-tagged Tbx3 and Flag-tagged GFP overexpressed in HEK293 cells. IP was performed with antibodies listed at top and immunoblotted for Tbx3. Myc-tagged Tbx3 co-IPs with Flag-tagged Kif7. Input lane was 5 s exposure (5" exp) while other lanes were 15 s (15" exp). (**E**) As in D, but in this case, blot probed for Kif7; confirms interaction of tagged, overexpressed proteins. (**F, G**) Representative images of anterior mesenchyme in sectioned forelimbs of control (**F**, *Tbx3fl/fl*) and mutant (**G**, *Tbx3fl/fl;PrxCre*) E10.5
*Figure 4 continued on next page*

*Figure 4 continued*

embryos stained for the ciliary marker Arl13b (red), Kif7 (green) and DAPI (DNA, blue). White arrowheads highlight cilia with multiple punctae or streak of ciliary Kif7 immunoreactivity (yellow) indicating translocation of Kif7 within the cilia. Please also see *Figure 5—source data 2* for z-stacks of additional Kif7 stained limb section. (H) Quantification of Kif7 staining pattern from multiple limb sections and three embryos of each genotype scored blinded to genotype. 10% fewer cilia have evidence of Kif7 translocation (multiple punctae or streak of Kif7 immunoreactivity) in *Tbx3^{fl/fl}*;*PrxCre* mutants. N=1785 and 1792 cilia scored in controls and mutants, respectively. * p<0.001 There was no difference in the number of Kif7- cilia. Insets show digital zoom of cilia with representative pattern used for scoring. (I) Immunoblot assaying for Kif7 and b tubulin loading control in control anterior (CA), mutant anterior (MA), control posterior (CP) and mutant posterior (MP) e10.5 forelimb bud lysates. This is representative of four such experiments. (J) Western blot assaying for Sufu protein and b tubulin loading control in eE10.5 forelimb buds. Like *Kif7* mutants,*Tbx3*;*PrxCre* mutants have increased Sufu protein. Sufu is increased 1.8 fold when normalized to loading control in this representative immunoblot; N=4. K) Western blot assaying for Spop protein and actin loading control in E10.5 forelimb buds. Spop is increased 1.5 fold when normalized to loading control in this representative immunoblot; N=3. Both Sufu and Spop protein levels are increased in mutant forelimbs although their transcript levels are unchanged (*Figure 3— figure supplement 4*).

The following source data and figure supplements are available for figure 4:

**Source data 1.** CZI file containing z-stack of E10.5 sectioned limb shown in *Figure 4—figure supplement 1*.

**Figure supplement 1.** Kif7 is also present in cytoplasm and nucleus.

**Figure supplement 2.** Anterior mesenchymal limb cilia are bigger in Tbx3;PrxCre mutants compared to controls.

not detect any difference in the amount of Kif7 protein in control versus mutant limb bud compartments by western blot (*Figure 4I*, representative of four4 separate experiments). Levels of *Kif7* mRNA in the anterior limb mesenchyme were unaffected by loss of Tbx3 (*Figure 3—figure supplement 4*). There was a decrease in the transcripts posterior compartment but as shown, the protein level was unchanged.

One feature of *Kif7^{-/-}* mutants is excess Sufu due to increased protein stability (*Hsu et al., 2011*). Transcript levels of *Sufu* were unchanged in mutant forelimbs (*Figure 3—figure supplement 4*, no difference was detected by microarray or RNA-Seq). However, increased amounts of Sufu (and Spop) protein were present in mutant limb buds (*Figure 4J,K*; increased 1.5 fold in both cases).

Humans and mice with *Kif7* mutations have abnormally long cilia because Kif7 reduces the rate of microtubule growth in the ciliary axoneme (*He et al., 2014*; *Putoux et al., 2011*). With 3D images obtained from 100X confocal z-stacks of Arl13b stained limb sections, we used the 3D object counter from ImageJ to calculate the volume and surface area to volume ratio to derive the length of cilia. Consistent with aberrant, but not absent, Kif7 function, we found that while the range of cilia volumes detected were the same between mutants and controls, the distribution was not: mutants have an increased fraction of larger cilia and an average volume 18% greater (*Figure 4—figure supplement 2A, B*; 475 and 575 cilia assayed in controls and mutants, respectively). Mutants and controls had superimposable surface area/volume ratios (*Figure 4—figure supplement 2C*), indicating that the shape of cilia was not different thus, the derived length was 6% greater in mutants.

Together, these findings indicate that loss of Tbx3 results in aberrant ciliary localization of Kif7 and are consistent with abnormal Kif7 function.

## Tbx3 is present in primary cilia and translocates in response to Hedgehog pathway activity

Kif7 and other proteins required for Gli3 processing and function are present in, or translocate to, primary cilia in response to Hedgehog pathway activity (*Goetz and Anderson, 2010*; *Ryan and Chiang, 2012*). Dual immunostaining for Tbx3 and Arl13b on whole mount optically sectioned E10.5 forelimbs shows that Tbx3 is present in control limb anterior mesenchymal cilia (*Figure 5 A-E*, *Figure 5—source data 1*). Specificity of Tbx3 staining was confirmed by loss of signal in mesenchymal cilia of *Tbx3*;*PrxCre* mutant limbs (*Figure 5F-J*, *Figure 5—source data 2*). The digital image overlap calculator in Zen software showed that 18/50 anterior mesenchymal cilia were Tbx3+ (36%, *Figure 5—figure supplement 1A,B*). Of note, no epithelial cilia were Tbx3+ in control limb epithelium, providing an internal negative control for the signal in mesenchymal cilia. This same calculation in *Tbx3*;*PrxCre* mutant anterior mesenchyme showed only 2/54 (<4%) of mesenchymal cilia had

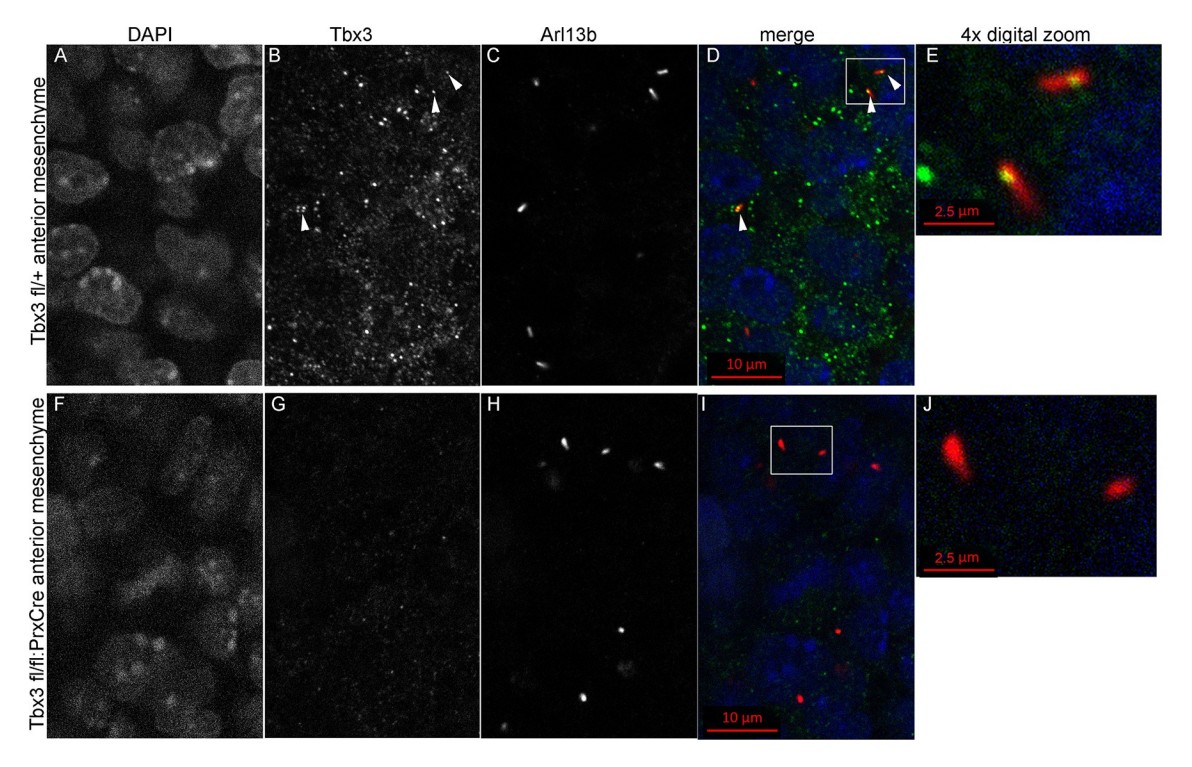

**Figure 5.** Tbx3 localizes to the primary cilia in limb mesenchyme. (**A–D**, **F–I**) Confocal 100X single Z-plane immunofluorescence images from optically sectioned E10.5 control (top panels **A–D**) and *Tbx3;PrxCre* (**F–I**) anterior limb buds after immunostaining with: Hoechst (DNA, blue), C-terminal anti-Tbx3 antibody (green, *Frank et al., 2013*), anti-Arl13b (red, cilia). Arrowheads demarcate Tbx3 colocalization with cilia marker. Panels E and J are further digital zooms of white boxed cells in D and I. The entire z-stacks containing these planes are in *Figure 5—source data 1,2*.
The following source data and figure supplements are available for figure 5:

**Source data 1.** Czi file of z-stack through the region of control anterior limb shown in *Figure 5A–E*.
**Source data 2.** Czi file of z-stack through region of mutant anterior limb shown in *Figure 5F–J*.
**Figure supplement 1.** Digital image overlap of Tbx3 and Arl13b in limb bud anterior mesenchyme.
**Figure supplement 2.** Tbx3 immunoreactivity in limb cilia is also detected by a commercial anti-Tbx3 antibody against the N-terminus of Tbx3.

background Tbx3 signal (*Figure 5—figure supplement 1, C, D*). These findings were reproduced with a commercially available anti-Tbx3 antibody (Abcam ab99302) which also showed Tbx3 ciliary staining on control limb mesenchymal cilia (24/87, 28%) that was virtually absent in *Tbx3* mutant limbs (2/52, <4%; *Figure 5—figure supplement 2*).

Murine embryonic fibroblasts (MEFs) are a robust system for studying ciliary proteins (*Chen et al., 2009*; *Dorn et al., 2012*; *Liem et al., 2012*; *Ocbina and Anderson, 2008*; *Rohatgi et al., 2007*), so we used them to further explore Tbx3 localization and trafficking. In untreated wild type MEFs, our custom C-terminal antibody detected Tbx3 in a subset of cilia (30%, N=56; *Figure 6A and B, a1-a5, b1-b5*; *Figure 6—source data 1,2*). No Tbx3+ cilia were detected in *Tbx3* null MEFS (*Figure 6C–F*; *Figure 6—source data 3*). Treatment of wild type MEFs with the smoothened agonist SAG increased the number of Tbx3+ cilia from 30% to 75% (*Figure 6G-J* p<0.005, *Figure 6—source data 4*, *Figure 6—figure supplement 1A*). Lysates from control and SAG-treated MEFs showed no detectable difference in Tbx3 protein levels indicating the increased Tbx3 signal in cilia was due to trafficking rather than increased protein levels (*Figure 6—figure supplement 1B*). The presence of Tbx3 in cilia and response to Hedgehog pathway activity were also detected with a commercially available anti-Tbx3 antibody with both SAG and Shh stimulation (*Figure 6—figure supplement 1C,*

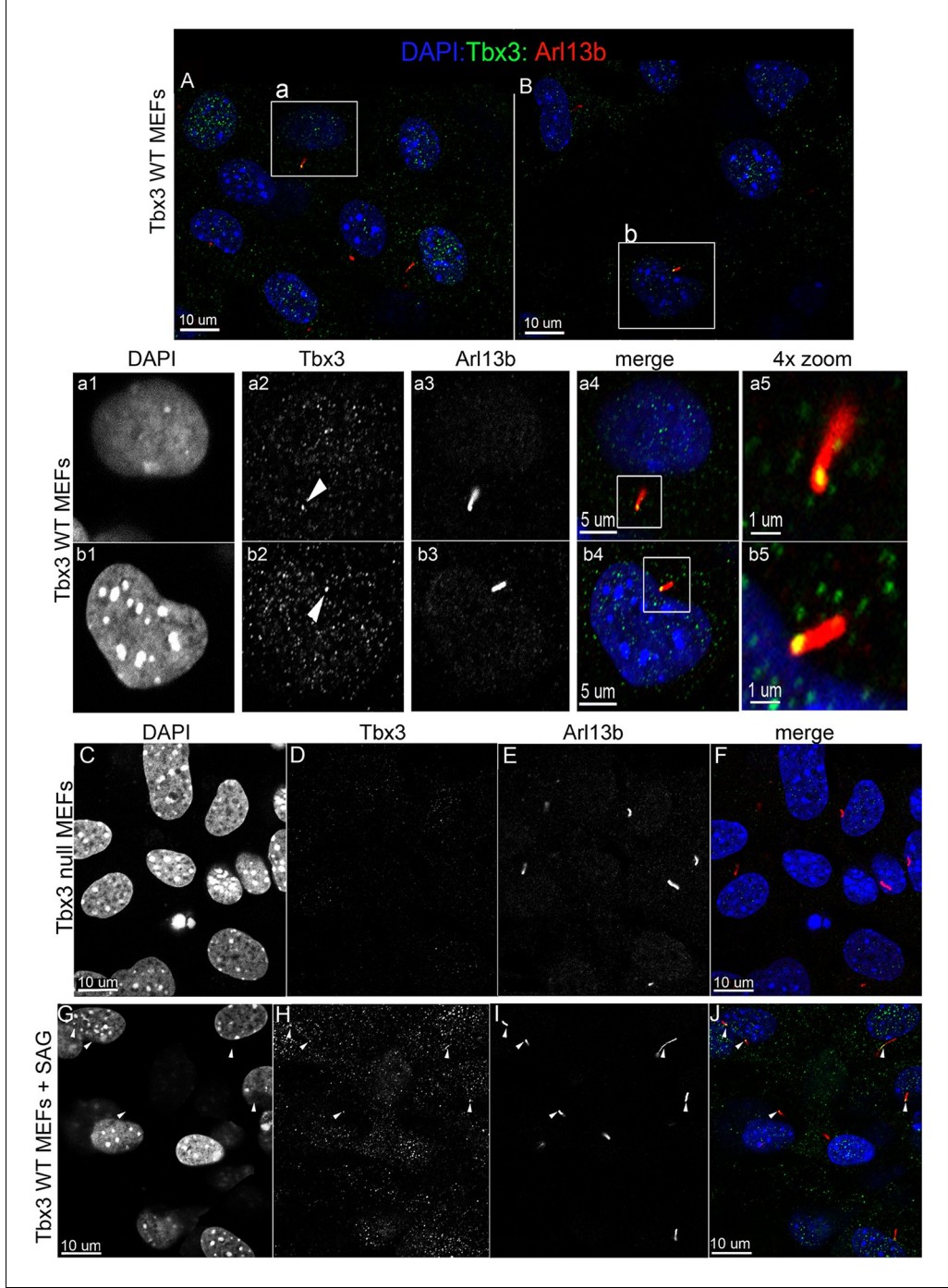

**Figure 6.** Tbx3 is present in some cilia at baseline in Murine Embryonic Fibroblasts and trafficks to cilia in response to hedgehog pathway activation. (**A**, **B**) Confocal, 100X single z-plane immunofluorescence images from two different fields of wild type MEFS after immunostaining for: DAPI (DNA, blue), Tbx3 (green, c-terminal anti-Tbx3 antibody; *Frank et al., 2013*), Arl13b (red, cilia). White boxed regions outline single cells that are shown at higher magnification in panels **a1–a5** and **b1–b5**. Please see *Figure 6—source data 1,2* for z-stacks. **a1–a4**, **b1–b4**) Single cells from white boxed areas in panels A and B. Individual cilia are shown in a5 and b5. White arrowheads highlight Tbx3+ cilia. (**C–F**) Tbx3 null MEFs show loss of Tbx3 immunoreactivity in cilia and other cellular locations. Please see *Figure 6—source data 3* for z-stack. (**G–J**) As in **A** and **B**, but MEFs were treated with smoothened agonist (SAG). White arrowheads highlight Tbx3+ cilia. Please see *Figure 6—source data 4* for z-stack.

*Figure 6 continued on next page*

*Figure 6 continued*

The following source data and figure supplement are available for figure 6:

**Source data 1.** Czi file showing z-stack of wild type MEFs imaged in *Figure 6* panel A.
**Source data 2.** Czi file showing z-stack of wild type MEFs imaged in *Figure 6* panel B.
**Source data 3.** Czi file showing z-stack of Tbx3 null MEFs imaged in *Figure 6* panel C–F.
**Source data 4.** Czi file showing z-stack of SAG treated MEFs imaged in *Figure 6* panel G–J.
**Source data 5.** Czi file showing z-stack of SAG treated MEFs imaged in *Figure 6—figure supplement 1* panel C.
**Source data 6.** Czi file showing z-stack of SHH treated MEFs imaged in *Figure 6—figure supplement 1* panel D.
**Figure supplement 1.** Tbx3 immunoreactivity in cilia increases in response to Hedgehog pathway stimulation without an overall increase in Tbx3 protein levels.

*D* and *Figure 6—source data 5,6*). In total, these data show that Tbx3 is present at baseline in cilia, and is trafficked to cilia in response to hedgehog signaling.

## Tbx3 co-localizes with Gli3 in the primary cilia

The presence of Tbx3 in cilia and the known association of Gli3 and Kif7 in primary cilia (*Endoh-Yamagami et al., 2009*) led us to test for interaction between endogenous Tbx3 and Gli3. Co-immunoprecipitation of E10.5 forelimb lysates showed that Tbx3 co-IPs with endogenous Gli3 (*Figure 7A*, and with Kif7 as shown previously), and that both Gli3FL and Gli3R are complexed with Tbx3 in the limb bud (*Figure 7B*, lane 1). These interactions are also detected in whole embryos (*Figure 7C*; specificity/efficiency of Gli3 and Kif7 IPs are shown in *Figure 7C' and C"*, respectively. Additional experiments showing this interaction are in *Figure 7—figure supplement 1*). Overexpression of Flag-tagged Gli3 and Myc-tagged Tbx3 in HEK293 cells followed by co-immunoprecipitation did not indicate direct interaction in this cellular context (*Figure 7—figure supplement 2*), suggesting that their association occurs within a larger complex that includes Kif7 (*Figure 4*).

Immunofluorescence assay of untreated MEFs for Tbx3, Gli3, and Arl13b by triple immunocytochemistry showed that 90% of cilia were Gli3+; Tbx3 colocalized with 66% of the Gli3 signals (N=38 total cilia scored, *Figure 7D–I' Figure 7—source data 1*). Treatment with SAG increased the fraction of Gli3+ cilia to >95% and all but two of those Gli3 signals colocalized with Tbx3 (N=34, *Figure 7J–O'*, *Figure 7—source data 2*). These results are quantified in *Figure 7P*.

## Loss of Tbx3 compromises the Sufu/Kif7 complex required for Gli3FL stability and normal processing of Gli3R

To obtain further mechanistic understanding into how loss of Tbx3 results in decreased Gli3 proteins, we assayed levels and interactions of members of the Gli3 processing machinery. Gli3FL stability and partial processing versus complete degradation are regulated by opposing actions of Suppressor of fused (Sufu) and speckle-type POZ protein (Spop), respectively (*Wang et al., 2010*; *Wen et al., 2010*). In the absence of Smoothened activation, mammalian Sufu recruits GSK3β to phosphorylate Gli3FL downstream of PKA, allowing for its partial processing to Gli3R (which requires Kif7 and intact cilia) (*Kise et al., 2009*; *Tempe et al., 2006*; *Wang et al., 2000*). Spop is an adaptor for Cullin3-based E3 ubiquitin ligase that drives complete degradation of Gli3FL in the absence of Sufu, but facilitates its processing to Gli3R when Sufu is present (*Humke et al., 2010*; *Wang et al., 2010*; *Zhang et al., 2006*). In contrast to Kif7 and Gli3, we did not detect interactions between Tbx3 and either Sufu or Spop (*Figure 7—figure supplement 3*). The observation that Gli3FL is virtually undetectable in *Tbx3;PrxCre* anterior mesenchyme (*Figure 4A'*) indicates that despite the increased amount of Sufu (*Figure 4H*, *Figure 8—figure supplement 1*), it fails to prevent degradation of Gli3FL in the absence of Tbx3.

We next tested whether decreased levels of Gli3 proteins in the anterior mesenchyme reflected perturbed interactions between Sufu, Gli3 and Kif7. Limitations in sample quantity made it unfeasible

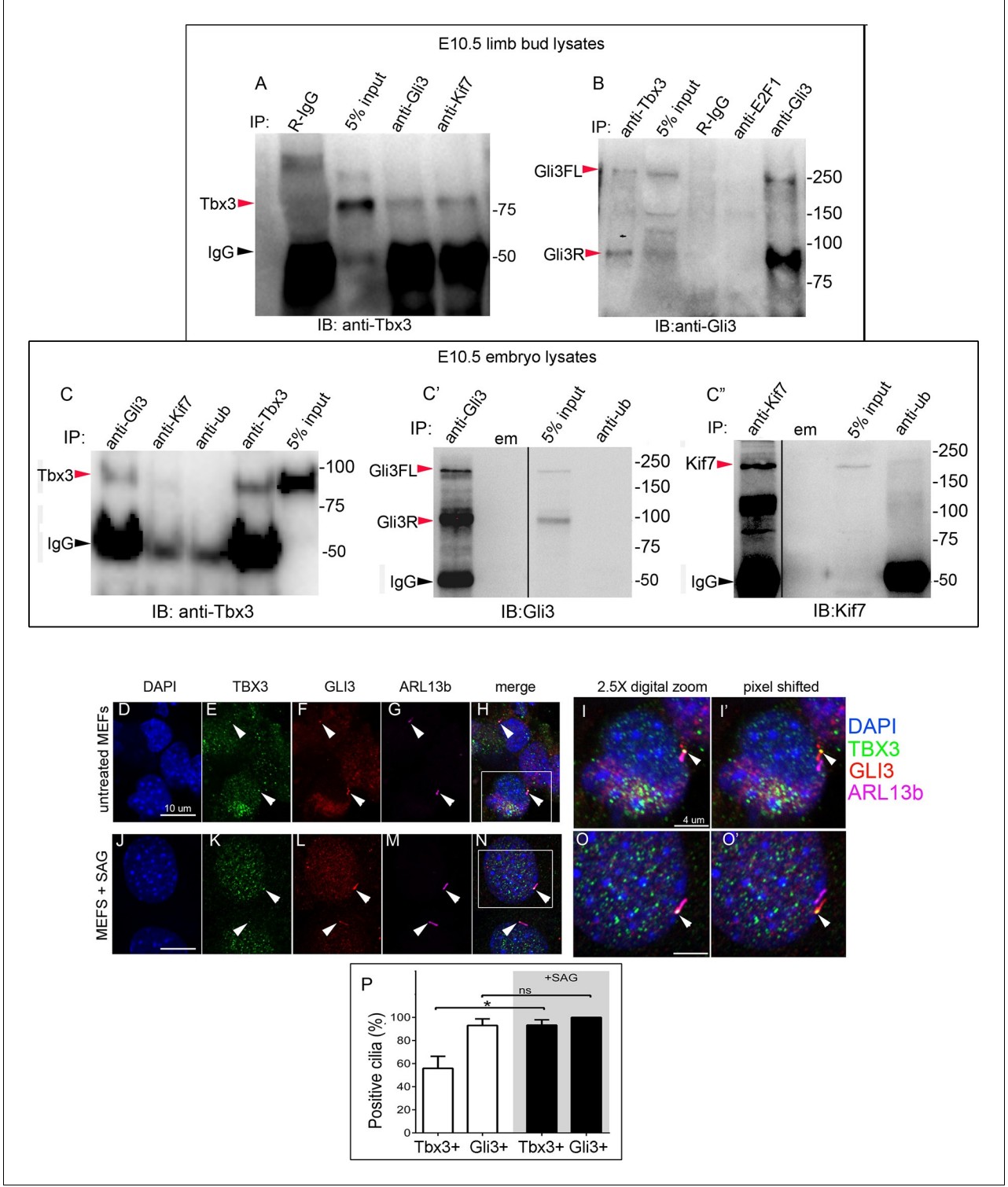

**Figure 7.** Tbx3 interacts with Gli3 in the limb bud and trafficks with Gli3 in primary cilia. (**A**, **B**) Immunoprecipitation (IP) of E10.5 forelimb bud protein lysates with antibodies listed at the top of panel and immunoblotted (IB) to detect Tbx3 (**A**) or Gli3 (**B**). Black arrowhead indicates IgG. Gli3FL and Gli3R (red arrowheads in B) both co-immunoprecipitate with Tbx3. (**C**) Immunoprecipitation (IP) of E10.5 whole embryo protein lysates with antibodies listed at the top of panel and immunoblotted to detect Tbx3. Tbx3 co-IPs with Gli3. Specificity and efficiency of anti-Gi3 and anti-Kif7 antibodies in whole embryo lysates tested are shown in panels **C'** and **C"**. Additional experiments demonstrating Tbx3/Gli3 interactions are in *Figure 7—figure supplement 1*. Em, empty lane (**D–I'**) Confocal 100X single Z-plane immunofluorescence images of vehicle (DMSO) treated MEFS after immunostaining for: DAPI (DNA, blue), Tbx3 (green, *Frank et al., 2013*), Gli3 (red), Arl13b (pink, cilia). Panel **H** is merged image of (**D–G**). Panel **I** is 2.5X digital zoom of

*Figure 7 continued on next page*

*Figure 7 continued*

the boxed cell in panel **H**, and **I'** shows the pink (cilia) channel pixel shifted to permit visualization of colocalized Tbx3 and Gli3 (yellow) within the cilia. White arrowheads highlight Gli3/Tbx3 colocalization. Please see *Figure 7—source data 1* for z-stacks. (**J–O'**) As above, but MEFS were treated with SAG in DMSO. Please see *Figure 7—source data 2* for z-stack. (**P**) Quantitation of Tbx3+ and Gli3+ cilia in MEFS -/+ SAG. SAG treatment causes the majority of cilia to become Tbx3+ and these ciliary Tbx3 signals all colocalize with Gli3.

The following source data and figure supplements are available for figure 7:

**Source data 1.** Czi file showing z-stack of wild type MEFs imaged in *Figure 7* panel D-I'.

**Source data 2.** Czi file showing z-stack of SAG treated MEFs imaged in *Figure 7* panel J-O'.

**Figure supplement 1.** Tbx3 and Gli3 coimmunoprecipitate in whole embryo protein lysates.

**Figure supplement 2.** Tagged Tbx3 does not co-IP with tagged Gli3 in HEK293 cells.

**Figure supplement 3.** Tbx3 does not co-IP with Sufu or Spop in mouse embryo lysates.

to perform multiple co-IPs on isolated anterior forelimb buds, so we assayed lysates from control and $Tbx3^{\Delta fl/\Delta fl}$ E10.5 embryos. The altered levels of protein expression seen in mutant limb buds were recapitulated in whole embryos (*Figure 8—figure supplement 1*). As seen in mutant limb buds, the amount of both Gli3FL and Gli3R protein is reduced in $Tbx3^{\Delta fl/\Delta fl}$ embryos (*Figure 8A*, lanes 1–4; *Figure 8—figure supplement 1*; *Figure 8—figure supplement 2A and B*). Notably, the interaction between Gli3 and Sufu is reproducibly decreased in excess of the decrement in Gli3 proteins; this is evident by quantitating and comparing the ratio of Gli3 proteins detected in control versus mutant to that complexed with Sufu. For example, in the representative experiment shown in *Figure 8A*, the Gli3 FL band intensity ratio is ~1.6 fold greater in controls (*Figure 8A*, lanes 1 and 3) than in mutants (*Figure 8A*, lanes 2, 4), while the amount of Gli3FL protein that was immunoprecipitated with Sufu is 4.6 fold greater in controls than mutants (*Figure 8A*, lane 9 versus 10). Mean band intensity ratios of 3 replicate experiments are shown in *Figure 8A'*. The decreased Gli3/Sufu interaction in the absence of Tbx3 was also evident when assayed in the opposite direction, that is, by IP of Gli3 and immunoblotting for Sufu (*Figure 8B*, panel B' shows relative band intensities for the experiment shown). Sufu/Gli3R interactions were also affected (*Figure 8A'*; *Figure 8—figure supplement 2A–B*). These findings indicate that normal stoichiometry of the interaction of Gli3 proteins with Sufu requires Tbx3.

In wild type embryos, we detected only trace interaction between Kif7 and Gli3R assaying by co-IP in either direction (*Figure 8C*, lane 3; *Figure 8D*, lane 7; *Figure 8—figure supplement 2C*, lane1). However, there was a robust and reproducible interaction in mutants despite the decrease in the total amount of Gli3 proteins. In the representative experiment shown in *Figure 8C*, in which we IP'd for Kif7 and assayed for Gli3, the band intensity ratios of control (lane 3) to mutant (lane 4) were 0.1 for Gli3FL and 0.25 for Gli3R. Quantification of the band intensity ratios from three replicate experiments in which we IP'd for Kif7 and assayed for Gli3 is shown in *Figure 8C'*. Furthermore, the increased interaction of Kif7 with Gli3 was confirmed by IP of Gli3 and assay for Kif7, as shown in *Figure 8D*: the band intensity ratio of control (lane 7) to mutant (lane 8) was 0.1, indicating a marked increase in interaction in mutants (*Figure 8D'* graphs relative band intensities in experiment 8D).

Lastly, the interaction between Sufu and Kif7 was reproducibly decreased when assayed by co-IP in either direction (*Figure 8E*, lane 3 versus 4; *Figure 8F*, lane 3 versus 4; *Figure 8—figure supplement 2D*, lane 7 versus 8). Quantification of the band intensity ratios from replicate experiments in which we IP'd for Sufu and assayed for Kif7 is shown in *Figure 8C'*. *Figure 8F'* graphs relative band intensities in experiment 8F in which we IP'd for Kif7 and assayed for Sufu. Note that this decrease in Sufu/Kif7 interaction occurs despite increased and normal levels of these proteins, respectively. In total, these data indicate that Tbx3 is required for normal stoichiometry and function of the Sufu/Kif7 complex that stabilizes and processes Gli3 as modeled in *Figure 9*.

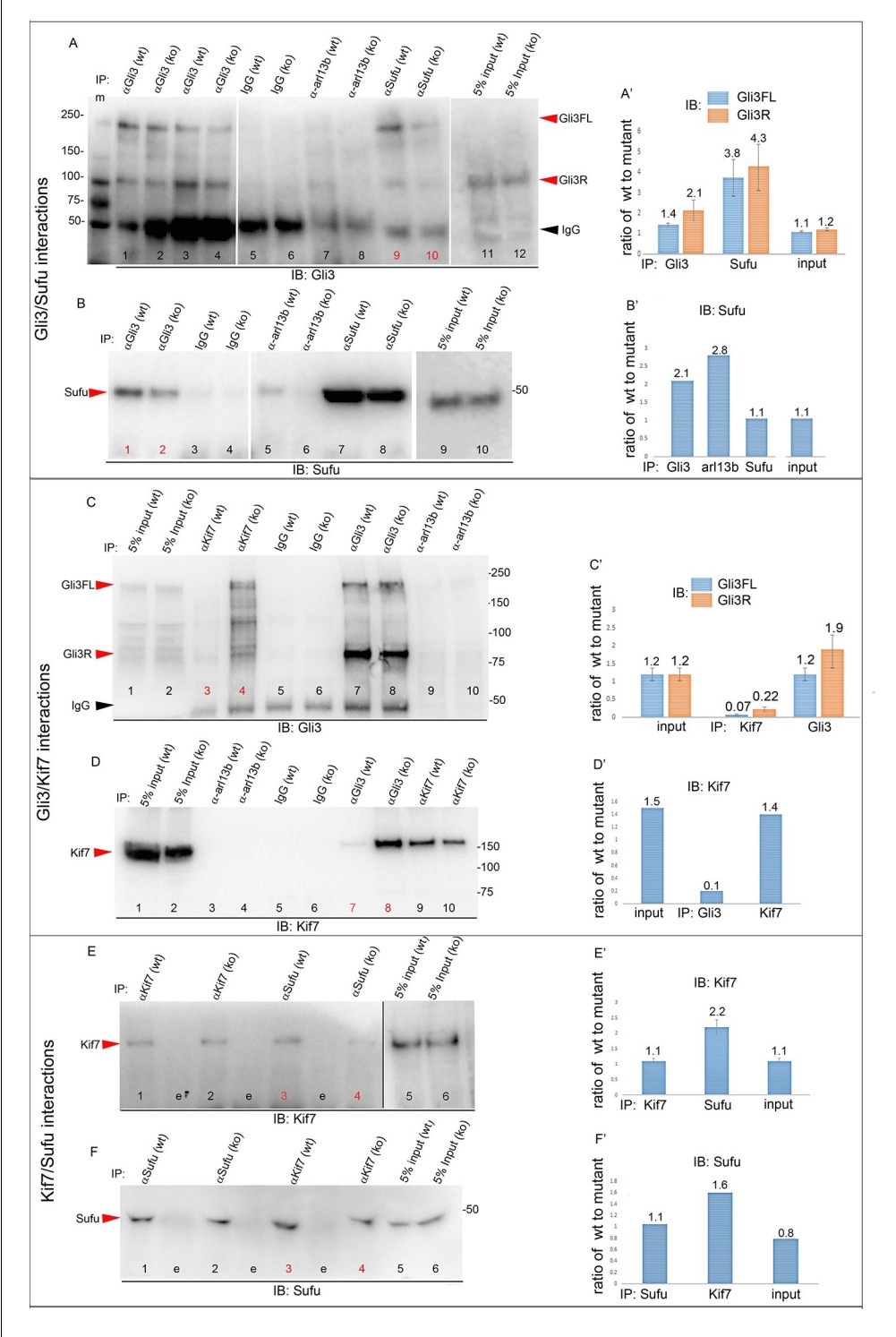

**Figure 8.** Altered stoichiometry of interactions between Gli3 and members of its processing/degradation complex. (**A**) Anti-Gli3 immunoblot (IB) on immunoprecipitates (IP) from antibodies listed at top on lysates from E10.5 control (wt) and $Tbx3^{\Delta fl/\Delta fl}$ (ko) embryos. Gli3FL and Gli3R are denoted by red arrowheads, IgG heavy chain with black arrowhead. Note decreased levels of IP'd Gli3FL and Gli3R in mutants compared with controls; the IPs in lanes 1–2 and 3–4 are two independent biologic replicates and the band intensity ratio of control to mutant for both GliFL and Gli3R was ~1.6. The interaction between Gli3 and Sufu is decreased more than can be explained by the overall decrement in Gli3 protein levels: in this representative experiment Sufu co-IPs 4.6X more Gli3FL in controls than in mutants (lane 9 versus 10). (**A'**) Bar graphs show the results of quantitation of band intensities from three3 replicate experiments measured with densitometry and presented as the ratio of signal detected in controls relative to mutants. (**B**) As in A but immunoblot probed for Sufu. Comparison of lanes 1 and 2 confirms decreased interaction between Gli3 and Sufu, despite preserved
*Figure 8 continued on next page*

*Figure 8 continued*

levels of Sufu in the mutants (lane 10). (**C**) Anti-Gli3 immunoblot with IPs as listed at top. Note increased interaction between Kif7 and Gli3 in mutants (lane 4), despite overall decreased level of Gli3 (lane 8). (**C'**) Quantitation of band intensities from three replicate experiments measured with densitometry and presented as a ratio of signal detected in control relative to mutant. Even though there is less total Gli3 protein, since there is increased interaction between Gli3 and Kif7 in mutants, the Kif7 co-IP control to mutant ratios are <1. (**D**) Anti-Kif7 immunoblot with IPs as listed at top. Confirms increased interaction between Gli3 and Kif7 in mutants. (**D'**) Quantitation of experiment in **D**. (**E**) Anti-Kif7 immunoblot with IPs as listed at top. The interaction of Kif7 and Sufu is decreased in the absence of Tbx3 (lane 4) despite preserved levels of both proteins (lane 6; *Figure 8*, panels **B**, **D** and **F**; *Figure 8—figure supplements 1* and *2*). (**E'**) Quantitation of band intensities from two replicate experiments measured with densitometry and presented as ratio of signal detected in control relative to mutant. There is 2 fold less interaction between Kif7 and Sufu in mutants. (**F**) Anti-Sufu immunoblot with IPs as listed at top. The interaction of Kif7 and Sufu is decreased in the absence of Tbx3 (lane 3 versus 4). (**F'**) Quantitation of findings in **F**.

The following figure supplements are available for figure 8:

**Figure supplement 1.** Altered protein levels observed in mutant limb buds are also apparent in whole embryos.

**Figure supplement 2.** Replicate experiments confirming altered stoichiometry of interactions between Gli3 and members of its processing complex in Tbx3 mutants.

## Discussion

### Role of Tbx3 in limb bud initiation

*Tbx3* expression in the LPM is required for normal *Tbx5* expression (*Figure 2A–C*) implicating *Tbx3* as one of the earliest limb initiation factors. This is consistent with our finding that Tbx3+ progenitors in the LPM give rise to the majority of limb bud mesenchyme. Determining the fate of these

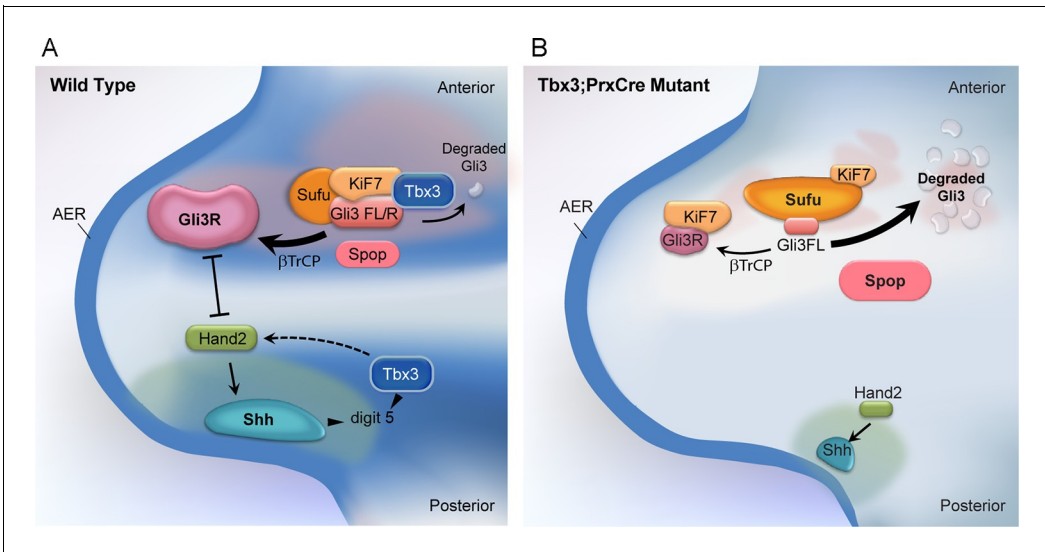

**Figure 9.** Model of compartment specific functions of Tbx3 in forelimb bud mesenchyme and altered interactions and stoichiometry of the Kif7/Sufu Gli3 processing complex in *Tbx3;PrxCre* mutants. In posterior forelimb mesenchyme, Tbx3 is required for normal levels of Hand2 upstream of Shh. Shh pathway activity and other Tbx3-reponsive factors promote digit 5 formation. In the absence of Tbx3, there is decreased expression of Hand2 and Shh and other digit 5 promoting pathways. In anterior mesenchyme, Tbx3 is in a complex with Gli3 proteins, Kif7 and Sufu and required for the stability of Gli3 FL and Gli3R. In the absence of Tbx3, Sufu and Spop protein levels are increased yet there is decreased interaction between Sufu and Kif7, and Sufu and Gli3. In mutant anterior mesenchyme, Gli3FL is barely detected: it is either degraded or converted to Gli3R. Levels of Gli3R are abnormally low due to a combination of decreased amount of Gli3FL precursor and its processing to Gli3R, and excess degradation. These findings are consistent with decreased function of cilia and Kif7 (required for processing of Gli3FL to Gli3R), and of Sufu (required for stability of both Gli3FL and Gli3R).

progenitors in the future will help reveal the mechanism for loss of ulna/fibula and digits 2–5 in *Tbx3* null embryos.

## Differential sensitivity of the left and right limbs to Tbx3

Intrinsic differences in anatomically bilaterally symmetric structures have long been suspected: acetazolamide teratogenizes only the right limb in rats (*Layton and Hallesy, 1965*; *Wilson et al., 1968*) and nitroheterocyclics such as valproate also induce unilateral defects (*Coakley and Brown, 1986*; *Fantel et al., 1986*). Directional asymmetry in limb size in human fetuses was recently reported (*Van Dongen et al., 2014*). Left/right identity differences in bilaterally symmetric structures such as the limbs may be upstream of secondary patterning events such as dorsal/ventral axis appropriate to body side. Shiratori and colleagues reported asymmetric expression of Pitx2c in the developing mouse limb and postulated functional left/right differences (*Shiratori et al., 2014*). Our finding that the left limb is more sensitive to Tbx3 than the right is consistent with this hypothesis and warrants further investigation in humans with UMS and other mouse models.

## Discrete functions for Tbx3 in the anterior and posterior limb bud mesenchyme

A model of molecular mechanisms contributing to the different anterior and posterior limb phenotypes of *Tbx3;PrxCre* mutants is shown in *Figure 9*.

Digit 5 formation is exquisitely sensitive to Shh activity and Grem1 (*Harfe et al., 2004*; *Scherz et al., 2007*; *Zhu and Mackem, 2011*; *Zhu et al., 2008*), so the altered expression of these genes in posterior mutant mesenchyme helps to explain loss of this digit. *Shh* null heterozygotes (*Shh*$^{+/-}$) have normal digit number, suggesting that the loss of digit 5 in *Tbx3;PrxCre* mutants is not solely due to decreased *Shh* expression and pathway activity in posterior mesenchyme. Importantly, it is not known if compensatory mechanisms preserve normal Shh protein levels or activity in *Shh*$^{+/-}$ mutants to support normal limb development. Indeed, a 'buffering system' to modulate polarizing activity by Shh was proposed in the chick limb (*Sanz-Ezquerro and Tickle, 2000*).

We found evidence of dysfunction of other digit 5 promoting pathways in posterior mesenchyme of *Tbx3;PrxCre* mutants. Compound *Hand2/Gli3* null mutants have more severe polydactyly than *Gli3* mutants (*Galli et al., 2010*), indicating that the role of Hand2 in digit formation is not limited to regulation of *Shh* expression. Aberrant expression of other BHLH factors (*Supplementary files 1,3*: *Hand1, Bhlhe40, Bhlhe41, Bhlha15*) may also contribute to the phenotype because the stoichiometry and interactions of BHLH factors have complex roles in limb development (*Firulli et al., 2005*). Altered levels of BMP responsive targets (*Dkk1, Noggin, Grem1, Grem2*) in *Tbx3;PrxCre* mutants suggest disrupted BMP signaling. Overexpression of *Gata6* decreases *Shh* and *Grem1* expression and causes loss of posterior digits (*Kozhemyakina et al., 2014*); we detected increased *Gata6* expression in the posterior mesenchyme of *Tbx3;PrxCre* mutants (*Supplementary file 3*). Additional studies are needed to determine if Tbx3 transcriptional or post-transcriptional functions directly regulate these pathways.

Although Tbx3 is also present in posterior mesenchymal cilia, we did not detect any evidence of altered stability or processing of Gli3 in the posterior mesenchyme (*Figure 4A*, representative of three replicates). Nonetheless, it is possible that decreased Shh signaling in the posterior of *Tbx3;PrxCre* mutant limb buds creates changes in Gli3 levels/ratios below our ability to detect. If so, decreased Shh activity would be predicted to increase Gli3R, decreasing digit number.

It is notable that decreased expression of *Tbx2* observed in *Tbx3;PrxCre* mutants would be predicted to result in increased *Grem1* expression (*Farin et al., 2013*) however, *Grem1* expression is decreased (*Figure 3*). Decreased Shh signaling and *Grem1* expression in the posterior mesenchyme would both be predicted to result in decreased *Fgf4* and *Fgf8* expression in the posterior AER (*Khokha et al., 2003*; *Michos et al., 2004*). Rather, the finding that these transcripts are increased in the posterior (*Figure 3—figure supplement 4*) indicates that loss of Tbx3 in posterior mesenchyme disrupts the Shh-Grem1-FGF signaling loop.

In wild type anterior mesenchyme, our data support the model that Tbx3 is part of a Kif7/Sufu complex that drives processing of the majority of Gli3FL to Gli3R and also prevents complete degradation of Gli3FL by Spop (*Wang et al., 2010*). Note that our model and data show a marked decrease in Gli3FL (virtually undetectable by western blot in anterior mesenchyme), but residual

Gli3R in mutants. This is consistent with the phenotype of Gli3 deficiency, and digit 1 polysyndactyly is also seen with decreased function of Kif7 or Sufu. Residual Gli3R in the mutants is sufficient to prevent the extreme polydactyly seen in complete absence of Gli3, Kif7 or Sufu. In *Tbx3;PrxCre* mutant limb buds, Kif7 and Sufu proteins are present at normal and increased levels, respectively, but their interaction with each other, and that of Sufu with Gli3, are decreased. Because Spop facilitates processing of Gli3FL to Gli3R in the presence of Sufu, increased levels of Spop in mutants drives complete degradation of any Gli3FL not converted to Gli3R, resulting in undetectable levels of Gli3FL in anterior mesenchyme. The decreased levels of Gli3R in the anterior likely result from a combination of decreased levels of Gli3FL precursor, inefficient processing due to altered Kif7/Sufu function, and excess Gli3R degradation, as evident by the lower molecular weight species present in *Figure 3A*.

This synthesis is consistent with the large body of published data indicating that anterior digit number is tightly regulated by the balance of Gli3A and Gli3R, in turn controlled by Kif7 and Sufu (*Cao et al., 2013*; *Wang et al., 2000*; *2007a*; *2007b*; *Zhulyn et al., 2014*). Kif7$^{-/-}$ and other ciliary mutants maintain robust levels of Gli3FL but have a marked decrease in processing it to Gli3R and their PPD phenotypes are consistent with decreased Gli3R (*Cheung et al., 2009*; *Endoh-Yamagami et al., 2009*). Furthermore, recent studies from Chi Chung Hui's group demonstrate that *Kif7;PrxCre* mutants have anterior PPD and that increasing Gli3R or decreasing Gli activators rescues all but digit 1 duplication (*Zhulyn and Hui, 2015*).

*Tbx3* mutant limbs also display evidence of Sufu dysfunction with markedly decreased levels of Gli3R and absent Gli3FL. Levels of Gli2, Gli3FL and Gli3R are all drastically reduced in *Sufu* mutants, even in the absence of cilia, consistent with the fact that Spop is not a ciliary protein and drives degradation of the full length proteins in the absence of Sufu (*Chen et al., 2009*; *Jia et al., 2009*; *Svard et al., 2006*; *Wang et al., 2010*). However, βTrCP and Spop can only mediate processing of Gli3FL to Gli3R in the presence of both Sufu and Kif7/intact cilia/intraflagellar transport (*Endoh-Yamagami et al., 2009*; *Law et al., 2012*; *Liu et al., 2005*; *Wang et al., 2010*; *Wen et al., 2010*). This processing is believed to occur at the basal body/centrosome (*Ryan and Chiang, 2012*; *Wang et al., 2013*; *Wen et al., 2010*; *Wigley et al., 1999*). Thus, our data strongly support that Tbx3 functions at the cilia or basal body, as a part of the complex that regulates Gli3 processing and stability (*Ryan and Chiang, 2012*; *Wen et al., 2010*).

Loss of Tbx3 could also influence stability and processing of Gli2. The anterior PPD phenotype of *Gli3* heterozygotes is slightly more severe in a *Gli2* null background (*Bowers et al., 2012*; *Mo et al., 1997*). Gli2 is also expressed in the posterior mesenchyme where it regulates digit patterning but not digit number (*Bowers et al., 2012*). Bowers' study also demonstrated that the role of Gli activators is in AP patterning of the posterior limb, whereas Gli repressors regulate digit number and anterior limb AP patterning. This is consistent with our findings that Tbx3 affects anterior digit number by regulating Gli3 repressor stability in the anterior mesenchyme.

The functional significance of increased interaction between Kif7 and Gli3R that we detect in mutants requires additional study nonetheless, since the anterior phenotype of *Tbx3;PrxCre* mutants is one of Gli3R deficiency rather than absence, we conclude that either there is a pool of Gli3R unbound by Kif7 and/or Gli3R complexed with Kif7 still has repressor function.

In addition to its function as a Gli3R co-repressor, Zic3 also regulates cilia morphogenesis and function (*Sutherland et al., 2013*). Since decreasing Zic3 levels in *Gli3*$^{+/-}$ mutants rescues their preaxial polydactyly (*Quinn et al., 2012*), *Zic3* overexpression in *Tbx3;PrxCre* mutants may contribute to the PPD phenotype.

Our data exclude ectopic Shh pathway activity as a cause of the PPD in *Tbx3;PrxCre* mutants: using multiple methods at multiple stages, we did not detect anterior *Shh*, *Gli1*, or *Ptch1* expression in the anterior mesenchyme. Notably, expression of the BMP antagonist *Grem1* is normally increased in PPD associated with ectopic anterior Shh pathway activity (*Lopez-Rios et al., 2012*; *Zhang et al., 2009*), but in *Tbx3* mutants, *Grem1* expression is markedly decreased. This is additional evidence in support of unique Tbx3-dependent pathways regulating digit number.

## Materials and methods

### Mice

Experiments were conducted in strict compliance with IACUC/AALAC standards. The $Tbx3^{fl}$ allele was detailed in *Frank et al. (2013)*. *Prx1Cre, RARCre* and $Rosa26^{LacZ}$ were previously reported (*Soriano, 1999*; *Moon and Capecchi, 2000*; *Logan et al., 2002*). Generation of the $Fgf8^{mcm}$ and $Tbx3^{mcm}$ alleles will be described elsewhere.

### β-Galactosidase detection

Males bearing $Fgf8^{MCM}$ or $Tbx3^{MCM}$ alleles were crossed with $Rosa26^{LacZ/+}$ females. Females were gavaged with tamoxifen (10mg/gm body weight) at stages stated in text. β−galactosidase activity was assayed using established protocols (*Park et al., 2006*).

### Skeletal preparations

E15.5 fetuses were fixed in 4% PFA overnight, rinsed in water for 2 days and alcian blue stained for 30 hr, then cleared in BABB. Older specimens were processed as in Moon et al. (2000).

### Whole mount RNA in situ hybridization

Digoxigenin-labeled riboprobes were generated according to manufacturer's instructions (Roche). Embryos were processed using a standard protocol (*Park et al., 2006*).

### RNA isolation and reverse transcription–qPCR analysis

Total RNA preparation, cDNA generation and qPCR were carried out as described in *Yu et al. (2010)*. Primer sequences are provided in *Supplementary file 5*. All qPCRs were performed on a minimum of three biologic replicates of pooled forelimb buds (*Figure 3*) or anterior and posterior forelimb bud segments (*Figure 3—figure supplement 4*).

### Gene expression analyses by microarray

Total RNA was prepared from three pools of dissected E10.25 control ($Tbx3^{fl/+}$) and $Tbx3;PrxCre$ mutant forelimbs using the RNAeasy Micro Kit (Qiagen 74004). The microarray and genomic analysis and bioinformatics core facilities at the University of Utah performed Agilent mouse whole-genome expression arrays and array image data analysis using Agilent Feature Extraction software. Subtle intensity-dependent bias was corrected with LOWESS normalization, with no background subtraction. Statistical analysis of normalized log-transformed data was performed in GeneSifter (www.genesifter.net). Differentially expressed transcripts were defined (adjusted for multiple testing using the Benjamini and Hochberg method) as p<0.05. The results presented in *Supplementary file 1* show transcripts that were statistically differentially expressed +/-1.3 fold in the mutant limb buds; yellow highlighting indicates changes that were replicated by RNA-Seq.

### RNA-sequencing to detect differential gene expression

Total RNA was isolated from pools of dissected anterior and posterior regions of E11 control ($Tbx3^{fl/+}$) and $Tbx3;PrxCre$ forelimbs using the RNAeasy Micro Kit (Qiagen). Each pool contained 12 forelimbs and two biologic duplicate pools were assayed. cDNA was generated, sequenced, and raw sequence reads were processed as described in *Kumar et al. (2014b)*. *Supplementary file 2* contains transcripts that are differentially expressed +/-1.3 fold (+/-0.38 in log base 2, column L) in control anterior (CA) compared to control posterior (CP) limb segments. The transcripts listed in *Supplementary file 3* were the result of mining the data to detect differential expression +/-1.3 fold (+/- 0.38 in log base 2, column L) in control posterior (CP) relative to mutant posterior (MP) segments. *Supplementary file 4* shows the result of mining the data to detect differential expression +/-1.3 fold (+/-0.38 in log base 2, column L) in control anterior (CA) relative to mutant anterior (MA) limb segments.

Note that transcripts that were not differentially expressed +/-1.3 fold are not listed on the tables. The complete unmined dataset is available on GEO.

## Immunoblotting

Protein lysates were prepared from E10.5 dissected limb buds or embryos or cultured MEFs using Dignam buffer. 50 ug of total protein were then subjected to SDS-PAGE analysis followed by immunoblotting according to standard protocols. Primary antibodies: Sufu (#2522, Cell signaling), Spop (#PA5-28522), Myc (#SC-789, Santa Cruz), Flag (#F7425, Sigma), E2F1 (#137415, Abcam), Ubiquitin (#7780, Abcam), Gli3 (AF3690, R and D systems), Tbx3 C-terminal antibody (*Frank et al., 2013*); Kif7 (ab 95884, Abcam); β tubulin (Santa Cruz).

## Immunoprecipitations

Immunoprecipitations were performed as described in *Kumar et al., 2014a* and *2014b*. Briefly, protein lysates were prepared from limb buds or embryos at E10.5 or transfected HEK293 cells using Dignam buffer C. Cleared protein lysates were obtained by centrifugation at 12,000g for 10 min. Equal amounts of protein lysates were incubated with 5–10μg of respective antibodies over night at 4°C with gentle shaking. Immune complexes were isolated with Protein-G Dynal beads and washed three times with Dignam buffer C. Precipitates were eluted from the beads by boiling in SDS-loading dye for 10 min, and analysed by western blotting by standard procedures using indicated antibodies at a dilution of 1:1000.

## Densitometry analysis

Immunoblot signals were quantified by densitometry using ImageJ64 software as per the procedure described by Luke Miller (http://www.lukemiller.org/ImageJ_gel_analysis.pdf).

## Immunofluorescence of sectioned and intact limb buds

Sections: Samples were fixed in 4% PFA at 4°C for 2 hr then washed with 0.3% Triton-X100 (in PBST). Heat retrieval in citrate buffer (Vector Laboratories) was performed for 2 min in a pressure cooker. Slides were washed with PBST and incubated in PBST with 5% serum corresponding to the secondary antibody origin for 1 hr. Slides were incubated with primary antibody in PBST 5% serum overnight at 4°C. Primary antibodies and dilutions: Tbx3 C-terminal antibody 1/200 (*Frank et al., 2013*), Tbx3 N- terminal antibody 1/100 (Abcam ab99302), Tbx3 internal epitope antibody (Santa Cruz A-20), Arl13b 1/100 (USDavis/NIH NeuroMab Clone N259B/66), Gli3 1/100 (R&D systems AF3690), pHH3 (Ser10), Millipore 06–570(1:2000); (1:50); Kif7 (1:200, kind gift from Dr. C-c Hui). After washing in PBST for 15 min, slides were incubated with secondary antibody from either Invitrogen or Jackson Immunoresearch diluted 1/1000 in PBST 2% BSA with Hoechst 33,342 at 1ug/ml for 1h at room temperature. Final wash was with PBST for 15 min and mounted using Fluoromount-G from Southern Biotech. Secondary antibodies: Donkey anti-rabbit Alexa 488; Donkey anti-goat Alexa 488; Donkey anti-mouse Alexa 647; Donkey anti-goat Alexa 594.

Whole forelimb buds: samples were fixed in 4% PFA for 1 hr at room temperature with PBST. Heat retrieval in citrate buffer (Vector Laboratories) was performed for 5 min at 100°C followed by washing with PBST. Limb buds were incubated in PBST with 5% serum that correspond to the secondary antibody origin (Goat or Donkey) for 2 hr. Limb buds were incubated with primary antibody in PBST with 5% serum overnight at 4°C, then washed with PBST once and incubated in PBST with 2% BSA for 4 hr. Incubated with secondary antibody and Hoechst as above with overnight at 4°C. Prior to imaging, samples were washed with PBST for 4 hrs and incubated in PBS/50% glycerol overnight.

TUNEL: was performed as described in (*Park et al., 2006*)

## Confocal microscopy and image processing of immunofluorescence on intact limb buds

Imaging was performed using a Zeiss confocal microscope LSM710 with Zen black imaging software http://www.zeiss.com/microscopy/en_us/downloads/zen.html. 100x objective was used.

To generate the Arl13b/Tbx3 colocalization maps, we used Zen to define pixels in each plane of z that exceeded an arbitrary threshold of 0.1 relative intensity in both Arl13b and Tbx3 channels using the imaging calculation subtab in the image processing menu. This was followed by calculation of the maximum intensity projection. To quantitate Tbx3 positive cilia, the 3D object counter in ImageJ

(*Bolte and Cordelieres, 2006*) was used employing the "redirection" option to superimpose Arl13b + objects into the Tbx3 channel.

## Assaying cilia volume and length

Cilia volume was assayed by quantitating Arl13b signal on 100x images of sectioned forelimbs from four pairs of mutant and control embryos stained for Arl13b and analyzing resulting in ImageJ. Imaged were Gaussian blurred using radius equal to '1' and further 3D object counter was used to measure volumes and surface area with threshold of 900 out of 4000 range. Distributions and parameters of the volume distributions shown in *Figure 4—figure supplement 2A and B* were calculated in Excel.

We calculated that the surface area and volume of both WT and mutant cilia tightly fit the equation: Surface=8.01 X Volume$^{0.69}$ (*Figure 4—figure supplement 2D*), which indicates that the cilia shapes are the same in the mutants and controls, and change size proportionally. In this case, the relative change in volume between mutant and control (Volume mutant)/(Volume control)=1.1747, which correspond to a change in length of 6%.

## MEF Immunocytochemistry

MEFs from E10.5 embryos were plated on fibronectin coated Matteks. MEFs were cultured and processed as in *Kumar et al. (2014a)*; cilia outgrowth was stimulated by culture in 0.5% FBS for 24 hr followed by incubation with 100 nM SAG or 100 nM SHH (Phoenix Pharmaceuticals) for 24 hr. Cells were washed 2x with PBS and fixed in 4% PFA for 20 min on ice, then washed 2X with PBS and permeabilized with 0.2% Triton X-100 in PBS for 10 min. After blocking in 5% donkey serum + 3% BSA for 1 hr, cells were incubated with anti-TBX3 (custom C-terminal or Santa Cruz sc-17871) and anti-Arl13b (1:50 NeuroMAb) at room temperature for 2 hr. Cells were washed 5 X 5 min in PBS and then incubated in secondary donkey anti-goat-488 or anti-rabbit 488 and donkey antimouse -594 (Jackson Immunoresearch). After washing 3 x 5min in PBS, cells were incubated in DAPI for 20 min. Cells were then washed 3X with PBS and mounted in Dabco for imaging. Images were captured on a Zeiss LSM-710 confocal microscope and processed using Zen software as described above.

## Assaying protein interactions with transfected, tagged proteins

HEK-293 cells were grown in DMEM supplemented with 10% fetal bovine serum (FBS) and penicillin/streptomycin. Cells were maintained in 5% C0$^2$ incubator at 37°C. Myc- tagged full length TBX3 was generated as previously described (*Kumar et al., 2014b*). The Flag-tagged Gli3 construct was obtained from Addgene (51246) and the Flag-tagged Kif7 construct was a kind gift of Kathryn Anderson. Transfections of plasmids were performed with Lipofectamine 2000 reagent (Invitrogen) as per the manufacturer's protocol.

## Acknowledgements

We thank Drs. Chi-Chung Hui, Cliff Tabin and Kathyrn Anderson for generously providing the Kif7 polyclonal antibody, the *Prx1Cre* mouse line, and the Flag-tagged Kif7 construct, respectively. We thank Susan Mackem, Steven Vokes and Brian Harfe for helpful discussions and Diana Lim for creating the graphic art. We thank Peter Gallagher for technical assistance.

## Additional information

### Author contributions

UE, PKP, TM, Acquisition of data, Analysis and interpretation of data; JMR, Conception and design, Acquisition of data, Analysis and interpretation of data; BM, AF, Acquisition of data, Analysis and interpretation of data, Drafting or revising the article; AMM, Conception and design, Acquisition of data, Analysis and interpretation of data, Drafting or revising the article, Contributed unpublished essential data or reagents

### Author ORCIDs

Anne M Moon, http://orcid.org/0000-0002-9090-1093

**Ethics**

Animal experimentation: This study was performed in strict accordance with the recommendations in the Guide for the Care and Use of Laboratory Animals of the National Institutes of Health. All of the animals were handled according to approved institutional animal care and use committee (IACUC) protocols with WCR IACUC protocol number 203-14.

# Additional files

**Supplementary files**

• Supplementary file 1. Differentially expressed transcripts detected by microarray of E10.25 control and *Tbx3;PrxCre* mutant forelimb buds. Table contains statistically significant differentially expressed genes determined as described in Methods section. Column 1 contains mean processed signal intensity of 3 biologic replicates from control limb, column 2 contains mean processed signal intensity of 3 biologic replicates from mutant limb. Fold changes are shown in Column E (Ratio). Yellow highlight of Gene ID (column N) indicates finding reproduced by RNA-Seq.

• Supplementary file 2. Differentially expressed transcripts detected by RNA-Seq of E11 control anterior forelimb buds versus control posterior forelimb buds. Transcripts that are differentially expressed +/- 1.3 fold (+/- 0.38 in log base 2, column L) based on mean FPKM values in control posterior (CP, column O) compared to control anterior (CA, column P) limb segments. Values for each biologic replicate are in columns Q-T.

• Supplementary file 3. Differentially expressed transcripts detected by RNA-Seq of E11 control posterior forelimb buds versus *Tbx3;PrxCre* posterior forelimb buds. Transcripts that are differentially expressed +/- 1.3 fold (+/- 0.38 in log base 2, column L) based on mean FPKM values in mutant posterior (MP, column O) compared to control posterior (CP, column P) limb segments. Values for each biologic replicate are in columns Q-T.

• Supplementary file 4. Differentially expressed transcripts detected by RNA-Seq of E11 control anterior forelimb buds versus *Tbx3;PrxCre* anterior forelimb buds. Transcripts that are differentially expressed +/- 1.3 fold (+/- 0.38 in log base 2, column L) based on mean FPKM values in mutant anterior (MA, column O) compared to control anterior (CA, column P) limb segments. Values for each biologic replicate are in columns Q-T.

• Supplementary file 5. qPCR primer sequences.

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
