## [Decision Letter]

[Editors’ note: this article was originally rejected after discussions between the reviewers, but the authors were invited to resubmit after an appeal against the decision.]

Thank you for choosing to send your work entitled "Tbx3 is a Ciliary Protein that Regulates Gli3 Stability to Control Digit Number" for consideration at *eLife*. Your full submission has been evaluated by Janet Rossant (Senior editor) and three peer reviewers, one of whom is a member of our Board of Reviewing Editors, and the decision was reached after discussions between the reviewers. Based on our discussions and the individual reviews below, we regret to inform you that your work will not be considered further for publication in *eLife*.

All referees agreed that the part of the paper describing the phenotype of Tbx3 loss in limb development is solid, and thus the comments concerning this part were relatively minor. However, the referees were much more skeptical about the strength of the experimental support for the most novel and interesting claim, the localization of Tbx3 to the cilia and the interaction of Tbx3 with the Hedgehog pathway components such as KIF7. Both the biochemical and subcellular localization experiments would need to be very significantly extended (for example, by using tagged proteins for immunoprecipitation assays), and more thorough controls for these experiments would need to be included. Also the strength and the value of the RNA-seq data presented in the paper were questioned, and these data thus need to be improved. Taken together, it appears that very significant additional work will be necessary to make the story convincing.

*Reviewer #1:*

This paper describes the role of Tbx3 in limb development and connects this function to cilia-dependent regulation of the Hedgehog pathway. The authors use conditional ablation of Tbx3 to characterize in detail its role in limb formation. The authors also show that Tbx3 localizes to cilia, interacts with Gli3 and Kif7 and affects the formation of protein complexes involved in Hedgehog signaling.

I think that the paper is interesting and potentially suitable for publication in e*Life*. However, while the phenotypic description of Tbx3 loss appears solid to me, the part describing the biochemical interactions and ciliary localization of Tbx3 misses important controls and is therefore less convincing.

1) Figure 5: it is very surprising that Tbx3 precipitates with equal efficiency with Tbx3 and Kif7 antibodies. Anti-Kif7 blot must be included here to show the efficiency of Kif7 immunoprecipitation. In the legend, the% of input loaded must be indicated, so that the efficiency of immunoprecipitation can be estimated.

A similar comment applies to Figure 6. A Tbx3 blot must be included in this panel, as well as the input lanes, so that the efficiency of immunoprecipitation can be estimated. The loading of IgG in the control lane seems to be lower than in the two other lanes, which makes the value of this control doubtful. It would be preferable, instead of IgG, to use as a negative control antibodies against another protein, which does not bind to Gli3 or Tbx3.

Similar comments also apply to the blots shown in Figure 7 and H. Arl13 blot should be shown in panel E, Kif7 blot in panel G and Sufu blot in panel H. Are the cut blots in panel E (including input lanes) derived from the same or different gels and do they show the same exposures? The observed differences in immunoprecipitation efficiency are not convincing and therefore need to be quantified using three independent experiments for all data shown in Figure 7.

2) Are there any indications that the interactions of Tbx3 with Kif7 and Gli3 might be direct? For example, do these proteins co-immunoprecipitate when overexpressed in cultured cells?

3) The specificity of the Tbx3 staining in cilia (Figure 6) must be confirmed using Tbx3 knockout cells.

*Reviewer #2:*

This is an extensive body of work using a range of genetic tools and molecular approaches to tease apart the function(s) of Tbx3 in the limb.

The first part of the paper describes an analysis of the phenotypes produced following conditional deletion of Tbx3 at different times and places during limb development using a variety of cre lines. I have relatively minor comments on this.

The second, and main part of the paper (Figure 5–Figure 8) covers all the work from which the title of the manuscript derive. I have several concerns here and in general remain skeptical (but with an open mind) about the data and particularly the interpretation.

In many instances I feel the authors have gone too far in their conclusions without the necessary experimental data to support the *final* conclusions.

First and foremost, immunostaining for Tbx3 and Arl13b on limb sections is interpreted as showing Tbx3 in limb mesenchymal cilia. Immunostaining of Tbx3 is throughout the entire cell and apparent co-expresssion with Arl13b could simply be explained by overlap in stain. Much higher resolution microscopy imaging that can circumvent these potential artifacts is required. The staining on MEFS do not clarify the issue. It is not clear why the Tbx3 staining pattern is so different in these cells from that shown in A-C. (Other issues such as the legend for panels H-K states 25/33 cilia were Tbx3+. I do not see more than 20 cells or so in this panel further compounding the problem) (It can also be argued that any patterns identified in MEFs may not have relevance to the limb.)

I also could not see a clear effect on KIf7 localization form the panels C in Figure 5. Table E appears to be an extreme way to show an apparent 8% difference on cilia length (I could not see where these numbers came from). This section ends with the conclusion that the data show that Tbx3 is present at baseline in cilia and is trafficked to and within cilia in response to hedgehog signaling. Unfortunately, I remain unconvinced of any of this.

There are similar issues with the data in Figure 7. Fundamentally I don't see clear evidence of co-expression of Tbx3, Gli3 Arl13b although I appreciate that IP data is also included.

The data on effect on GLi3 stability are potentially interesting but perplexing

What is the effect on Gli3 expression in the mutant limbs? It might be expected following reduction of hand2 and Shh expression in the mutant that Gli3 expression would be expanded but the western data would suggest not. Is there more GLi3FLin the MP? It is perhaps surprising that apparently equivalent levels of Gli3R are present in CP and MP.

The Tbx3 cKO is close (but I would not say identical to or a phenocopy of) a Het *Gli3* mutant. This is not conveyed in the final schematic which suggests a critical role of Tbx3 in Gli processing and that this in its absence *all* Gli processing is affected. I would expect a more profound polydactyly if this were the case. The actual 'polydactyly that presents lacks a structure that could be considered a true digit (1H, H').

A further question raised is that if Tbx3 is playing such a critical role in Gli processing/Shh signalling in the anterior is it or why is it not playing a similar role in the posterior. This is not discussed and does not appear to have been studied.

Some other minor issues from Figure 1 include:

1L: I do not see any loss of Sox9 activity in the region indicated with the red arrow head. This domain looks relatively normal to me.

I do not see clear evidence of a delay in ossification. In the text the authors conclude that this "reveals a role for Tbx3 in later aspects of Indian hedgehog regulated bone morphogenesis.” This has not been investigated to any extent and this type of comments goes far beyond what is actually shown in this paper. Also here the loss of the deltoid tuberosity need not have anything to do a direct effect on bone (cf Blitz, E, 2009).

Comparison of the phenotypes produced using the early acting (RARCre) and later acting (Prx1Cre) appears to show an effect of Tbx3 on the early stages of limb initiation that can be distinguished form later events of limb bud patterning through establishment of Shh in the posterior. A regulatory relationship between Tbx3, Hand2 and Shh has been reported in chick (Rallis, 2005).

Decreased expression of markers detected by RNAseq analysis is interpreted as evidence that fewer cells are undergoing condensation and chondrogenesis in posterior mesenchyme. This would be much better analyzing in a more direct way rather than making inferences from RNA-seq data.

Similarly, 5th digits agenesis is attributed to deceased cell number as opposed to decreased proliferation of digit 5 progenitors. These conclusions are drawn from a broad analysis of effects throughout the limb bud rather than focusing on the cells in question themselves.

The authors have some interesting results and observations. I remain open to the ideas they are suggesting but I do feel strongly that currently the data shown does not support their main conclusions.

*Reviewer #3:*

The study described the limb phenotype in several Tbx3 conditional and null mutants in careful detail. Most of the conclusions based on whole mount RNA in situs are convincing and new findings. The most novel and surprising finding of the study is that the transcription factor TBX3 is localized to the cilium and interacts with KIF7. HOX proteins have been shown to interact with and inhibit GLI3 in the limb, however cilium localization was not explored. Given that the vast majority of TBX3 protein is in the nucleus (Figure 6), it seems surprising to me that the tiny amount in the cilium can be detected with IP (Figure 7). The interaction with GLI3 could be in the nucleus and/or cilium, but KIF7 should not be in the nucleus. To further bolster the most significant claim of the paper, I think that additional experiments are needed. In particular, the specificity of the antibodies must be proven (e.g. using Tbx2 mutant MEFs for the Tbx3 antibody) and ideally some results confirmed using tagged proteins. Tagged GLI3 is available and other proteins could be easily generated. A genetic test would be to determine whether heterozygosity for Kif7 in mice (or knockdown in cells) rescues the defect proposed to be due to increased KIF7. In addition, many of the claims of changes in gene expression based on the RNA-seq data in Tables 2 and 3 do not seem to be significant differences, and some are opposite to what is said in the Results. Thus, the value of the RNA-seq data, is not clear, especially as only two samples were analyzed for each genotype, although I appreciate it is a difficult experiment. Many of the additional points listed below are simple changes that would make the paper more accessible and data more convincing.

Figure 3': The mutant limb looks smaller than the control limb in (A), therefore the domain is expected to be smaller. The text implies the domain is smaller relative to the size of the limb. Which is the case?

"[…]downregulation of posterior genes including Shh, Shh pathway members, Tbx2, Sall1, Dkk1, and Osr1[…]": The decrease is very little (e.g. Sall1 -0.3). More important, why are Grem1, Ptch1 and Gli1 not down in the RNA-seq, which would confirm the data in Figure 3 and address the question of whether the decrease is just due to the limb being smaller?

"[…]up regulation of (AFP, Ttr, Apob, Apoa1, Apoa4, Trf, Ttpa)[…](Table 2).”: Apob and Ttpa are actually down and the others are barely up in the anterior (Table 2).

Results: "more single puncta cilia in mutants than controls (Figure 5 84% vs 71%, p<0.001). The levels of Kif7 mRNA[…] unaffected[…] Tables 2, 3[…]": In Table 3 Kif7 is -0.3, which is greater than many of the changes in expression level claimed elsewhere in the text.

Results: "The presence of Tbx3 in cilia and response to Hedgehog pathway activation was confirmed with a commercially available anti-Tbx3 antibody and both SAG and Shh stimulation (Figure 6).": Tbx mutant fibroblasts should be shown to demonstrate specificity of the antibody.

Tagged proteins should be used to confirm the interactions thought to be detected with commercial antibodies.

Figure 7: quantification is needed.

Results: "Consistent with our hypothesis, there was decreased interaction between Gli3FL and Sufu in mutants (Figure 6, lane 8[…]": Do the authors mean Figure 7? Also, since Gli3 FL and R are reduced, does this experiment say anything?

Discussion: "Future studies will determine if Tbx3 transcriptional or posttranscriptional activity regulates production of a factor that represses expression or stability of Hand2 mRNA.": In the posterior limb, wouldn't Tbx3 also interfere with processing of Gli2/3 into activators, like in cilia mutants, and this could explain the posterior limb phenotype by the same mechanism as the anterior limb bud?

[Editors’ note: what now follows is the decision letter after the authors submitted for further consideration.]

Thank you for resubmitting your work entitled "Tbx3 is a Ciliary Protein and Regulates Gli3 Stability to Control Digit Number" for further consideration at *eLife*. Your article has been favorably evaluated by Janet Rossant (Senior editor) and three reviewers, one of whom is a member of our Board of Reviewing Editors. The manuscript has been improved but there are some remaining issues that need to be addressed before acceptance, as outlined below:

1) There are some remaining concerns about the quality of the biochemical data. Specifically, the composition and the order of loading of the lanes is not the same in Figure 4 and Figure 7, creating the impression that the shown immunoprecipitations are not from the same experiment. Gli3 and KIF7 blots should be included in Figure 7. Most importantly, error bars should be included in the plots showing the quantification of immunoprecipitation results in Figure 8 in order to illustrate that the experiment was performed more than once and the observed differences are significant.

2) Better discussion should be included to explain why the authors insist that TBX3 does not regulate GLI3 function in the posterior limb, and why they have ignored the highly related protein GLI2. If their hypothesis is correct, one would expect the same might be true for GLI2 since it also binds SuFu and requires cilia for proper processing. Also, if whole embryos reveal the same protein interactions as the anterior limb, why would the posterior limb be different? Gli2/3 double limb mutants could well have a worse posterior limb phenotype than Gli3 null or het mutants. Only a relatively late double conditional mutant of Gli2/3 has been published, and it indicated the posterior phenotype is worse than in single Gli3 mutants. Thus, some of the posterior phenotype in Tbx3 mutants could be due to the altered GLI processing and activity in this tissue, rather than just the decreased Shh expression, especially as Shh heterozygous mutants have normal limbs. Finally, it seems to be an over-interpretation of the data to say that the posterior limb is not dependent on cilia. In cilia mutants’ production of both activators and repressors is altered (diminished), but obvious phenotypes are only seen where the balance is greatly skewed. It would be nice to extend the Discussion of your paper to address these points.

[Editors' note: further revisions were requested prior to acceptance, as described below.]

Thank you for resubmitting your work entitled "T-box 3 is a Ciliary Protein and Regulates Gli3 Stability to Control Digit Number" for further consideration at *eLife*. Your revised article has been favorably evaluated by Janet Rossant (Senior editor) and a reviewing editor. The manuscript has been improved but there are some remaining issues that need to be addressed before acceptance, as outlined below:

1) The scan of the Western blot shown in Figure 7 is not publication quality. Please substitute it for a high-resolution scan. Also, please make sure that in all cases where lanes of Western blots have been left out, a clear separator line is present (this currently seems not to be the case in Figure 7 and Figure 8).

2. Please reconsider the title of your paper. "T-box3 is a ciliary protein and regulates.." doesn't read well. Furthermore, e*Life* encourages the authors to provide brief explanations for the acronyms used in the title and Abstract. For your paper, mentioning that Gli3 is a player in Sonic Hedgehog signaling would be appropriate.

---

## [Author Response]

[Editors’ note: the author responses to the first round of peer review follow.]

*Reviewer #1:*

*1) Figure 5: it is very surprising that Tbx3 precipitates with equal efficiency with Tbx3 and Kif7 antibodies. Anti-Kif7 blot must be included here to show the efficiency of Kif7 immunoprecipitation.*

We agree; we have repeated the co-IP a total of 4 times: a different experiment is shown (now in Figure 4) than in the original submission and another example is in the new Figure 7. The anti-Kif7 blot requested by the reviewer reveals that as predicted, Tbx3 is not as efficiently IP’d by anti-Kif7 antibody as Kif7 itself.

*In the legend, the% of input loaded must be indicated, so that the efficiency of immunoprecipitation can be estimated.*

% input added to the figures and legends as requested.

*A similar comment applies to Figure 6. A Tbx3 blot must be included in this panel, as well as the input lanes, so that the efficiency of immunoprecipitation can be estimated. The loading of IgG in the control lane seems to be lower than in the two other lanes, which makes the value of this control doubtful. It would be preferable, instead of IgG, to use as a negative control antibodies against another protein, which does not bind to Gli3 or Tbx3.*

We have made the additions and changes as requested, and used an additional negative control antibody as suggested by the reviewer. These data are now presented in the current Figure 7.

*Similar comments also apply to the blots shown in Figure 7 and H. Arl13 blot should be shown in panel E, Kif7 blot in panel G and Sufu blot in panel H. Are the cut blots in panel E (including input lanes) derived from the same or different gels and do they show the same exposures? The observed differences in immunoprecipitation efficiency are not convincing and therefore need to be quantified using three independent experiments for all data shown in Figure 7.*

Because it requires that a much greater amount of protein be loaded to detect these proteins from non- IP’d material, the blots in panel E were from a different gel to avoid exposure and well distortion issues that arise from the different amounts of protein loaded. The experiments originally presented in Figure 7 have now been repeated a minimum of 3 times and examples are shown in Figure 8 and Figure 8—figure supplement 2.

*2) Are there any indications that the interactions of Tbx3 with Kif7 and Gli3 might be direct? For example, do these proteins co-immunoprecipitate when overexpressed in cultured cells?*

We have performed the co-IPs with tagged Kif7 and tagged Gli3 as requested (Figure 4 and Figure 7—figure supplement 1) in HEK293 cells. We found that Flag-Kif7 and Myc-Tbx3 interact, but Flag-Gli3 and Myc-Tbx3 do not under these conditions.

*3) The specificity of the Tbx3 staining in cilia (Figure 6) must be confirmed using Tbx3 knockout cells.*

We agree and apologize for not including this critical control in our initial submission. We have now repeated these experiments and used *Tbx3* mutant limb buds (Figure 5) as well as *Tbx3* mutant MEFs (Figure 6) and confirmed the specificity of signal in the cilia. Please note that in addition to our custom antibody against the C-terminus of Tbx3, 2 different commercial anti-Tbx3 antibodies detected Tbx3 in cilia (Figure 5—figure supplement 2, Abcam anti-Tbx3 antibody; Figure 6—figure supplement 1, panels C, D SantaCruz A-20 anti-Tbx3 antibody).

*Reviewer #2:*

*First and foremost, immunostaining for Tbx3 and Arl13b on limb sections is interpreted as showing Tbx3 in limb mesenchymal cilia. Immunostaining of Tbx3 is throughout the entire cell and apparent co-expresssion with Arl13b could simply be explained by overlap in stain. Much higher resolution microscopy imaging that can circumvent these potential artifacts is required. The staining on MEFS do not clarify the issue.*

We agree; please see response to Reviewer 1 above and the new data provided in control and mutant limb buds, including 100X confocal maximum image projections in Figure 5 and the z-stacks provided in Videos 3 and 4.

*It is not clear why the Tbx3 staining pattern is so different in these cells from that shown in A-C.*

We have shown that the staining pattern of Tbx3 is very context/tissue dependent (Frank et al., 2012; Frank et al., 2013, Kumar et al., 2014a *eLife*). In this case, serum starved MEFs are very different than the rapidly proliferating mesenchymal cells in the limb bud. Even within the limb, cells in the AER have a different pattern of Tbx3 staining (mostly cytoplasmic, none in cilia) than those in the mesenchyme (cytoplasmic, nuclear and cilia). In some cell types, such as the dorsal root ganglia, all staining is nuclear. The MEF pattern shown here is very similar to what we demonstrated in MEFs in Kumar et al., 2014a, *eLife*.

*(Other issues such as the legend for panels H-K states 25/33 cilia were Tbx3+. I do not see more than 20 cells or so in this panel further compounding the problem) (It can also be argued that any patterns identified in MEFs may not have relevance to the limb.)*

We apologize for this error; the text was written to describe a larger, lower magnification panel that we subsequently chose to replace with the panel presented. In the revision, we have replaced all of this data with cleaner, higher magnification/better resolution confocal images in Figure 6 and Figure 7 that can also be viewed orthogonally and through z-stacks using the free Zen software that we employed to obtain and process the images http://www.zeiss.com/microscopy/en_de/downloads/zen.html. In addition to viewing z-stacks through the limb or MEFs, this software also permits the entire stack to be viewed in 3D and orthogonally so that the overlap is visible at the pixel level. We have also employed the image calculator function of this software to objectively quantitate overlap at the pixel level between the Arl13b and Tbx3 signals in control versus mutant. This method is the most sensitive and objective available to us to define true colocalization. It confirms the presence of Tbx3 in cilia and clearly shows Tbx3 immunoreactivity in limb mesenchymal cilia that is virtually absent after Cre recombination in mutants.

*I also could not see a clear effect on KIf7 localization form the panels C in Figure 5.*

The goal of the panel as shown was to provide the overall picture of a 40X confocal field and to show that there is not an appreciable difference in the number of ciliated limb mesenchymal cells between control and mutant. However, as stated in the legend, the white arrowheads denote the cilia with multiple spots or streak of Kif7 signal. This is quantitated on a much larger scale from many such fields in the bar graph which represents the results from 3 independent reviewers blinded to genotype, scored the Kif7 immunoreactive pattern of the cilia on 20 fields from 3 mutants and 3 controls.

*Table E appears to be an extreme way to show an apparent 8% difference on cilia length (I could not see where these numbers came from).*

We have replaced this Table with more accurate quantitation based on 3D analysis of the cilia, as described in response to Reviewer 1 above; this is in Figure 4—figure supplement 2.

This section ends with the conclusion that the data show that Tbx3 is present at baseline in cilia and is trafficked to and within cilia in response to hedgehog signaling. Unfortunately, I remain unconvinced of any of this.

*There are similar issues with the data in Figure 7. Fundamentally I don't see clear evidence of co-expression of Tbx3, Gli3 Arl13b although I appreciate that IP data is also included.*

Please see response to Reviewer 1 and extensive new, appropriately controlled data in Figure 5–Figure 8. We are confident that the reproducible immunofluorescence and IP data we have now provided address the reviewer’s doubts.

*The data on effect on GLi3 stability are potentially interesting but perplexing.*

We agree that this is an extremely interesting finding and while discovering the exact mechanism whereby Tbx3 stabilizes Gli3 is beyond the scope of this manuscript, the fact that Tbx3 is in a complex with Kif7 and Sufu which regulates processing and stability of Gli3 provides the first step toward understanding this.

*What is the effect on Gli3 expression in the mutant limbs? It might be expected following reduction of hand2 and Shh expression in the mutant that Gli3 expression would be expanded but the western data would suggest not. Is there more GLi3FLin the MP? It is perhaps surprising that apparently equivalent levels of Gli3R are present in CP and MP.*

As shown by the graph of qPCR results in Figure 3, and the new in situ images provided there Gli3 mRNA levels are increased, as is expected with decreased Shh activity (Wang et al.,2000). We have now included an early microarrary experiment which also showed upregulation of *Gli3* expression in mutants (new Table 2).

In the posterior mesenchyme, the level of Gli3R is very low compared to anterior and this has been published by numerous groups including (Wang et al.,2000). Although Shh signaling is reduced in *Tbx3* mutants, the westerns all indicated that baseline processing of Gli3FL to Gli3R is intact in mutant posterior mesenchyme.

The Tbx3 cKO is close (but I would not say identical to or a phenocopy of) a Het Gli3 mutant. This is not conveyed in the final schematic which suggests a critical role of Tbx3 in Gli processing and that this in its absence all Gli processing is affected. I would expect a more profound polydactyly if this were the case. The actual 'polydactyly that presents lacks a structure that could be considered a true digit (1H, H').

The phenotype of Gli3 null heterozygotes or Gli3R deficiency varies depending on genetic background and protein level; and others have also reported polysyndactyly of digit 1 (Wang 2007;Hill et al., 2009; Lopez-Rios et al., 2012;Quinn et al., 2012). The polysyndactylous digit of Tbx3;PrxCre mutants has two phalanges, the primary criteria for digit 1 (thumb) identity. Even in our mutants, although the penetrance is 100%, the severity of the duplication varies. *Tbx3;PrxCre* mutants are Gli3 deficient and not completely lacking Gli3R, thus PPD is the observed (and expected) phenotype. In the Discussion we consider other molecular events occurring in these mutants that may contribute to the phenotype, including overexpression of Zic3.

Our schematic model shows that in the anterior mesenchyme, some Gli3FL remains complexed with Sufu, but most is processed to Gli3R (which is also the case in controls); some of this Gli3R is aberrantly degraded which is what our data show. Our model does *not* indicate that all Gli3 processing is affected. The very low levels of Gli3FL detectable in the mutants are consistent with excess degradation of Gli3FL.

*A further question raised is that if Tbx3 is playing such a critical role in Gli processing/Shh signalling in the anterior is it or why is it not playing a similar role in the posterior. This is not discussed and does not appear to have been studied.*

With due respect to the reviewer, this is not discussed because, unlike other regions of the embryo, Shh signaling in the posterior limb mesenchyme is not dependent on cilia or Gli3R. Mouse mutants of Kif7 and ciliary components have anterior PPD; posterior digits are apparently normal. Our data indicate that Tbx3 regulates ligand-dependent Shh signaling in the posterior mesenchyme by affecting the level of *Hand2* expression upstream of *Shh* and its downstream activator functions. In the anterior, we show strong evidence that Tbx3 regulates ligand-independent pathway function by controlling the stability and processing of Gli3. Gli3 processing requires Kif7, Sufu and intact cilia. Although the interactions and processing may not occur within the cilia proper; the literature supports that they occur in the cytoplasm or centrosome/basal body adjacent to the cilia (reviewed in Ryan and Chiang 2012; Ingham and McMahon 2009; and Wang et al., 2013; Wen et al., 2010, Humke et al., 2005; Tukachinsky et al., 2010).

As the reviewers are aware, the cellular compartments in which different functionally relevant interactions are occurring is an ongoing and highly controversial area in the field for all components of the pathway. Elegant work has been done with tagged proteins and overexpression systems. Despite this, these issues have not yet been resolved, especially for endogenous interactions and functions of Kif7, Sufu or Gli3. What is clear is that even if the functionally relevant interactions are occurring in the cytoplasm adjacent to (or even far) from cilia, processing of Gli3 to Gli3R requires the cilia.

*Some other minor issues from Figure 1 include:*

*1L: I do not see any loss of Sox9 activity in the region indicated with the red arrow head. This domain looks relatively normal to me.*

We agree that *Sox9* expression at E10.5 appears normal however, the morphology of the limb is not at this stage and is variable as is apparent from the images in Figure 3 as well. In Figure 1, the red arrow denotes abnormal indentation/lack of posterior tissue while the red bracket highlights extra tissue in the anterior, digit 1 forming region. We have clarified the text on this point.

*I do not see clear evidence of a delay in ossification.*

We have now added a label for the ossification center (oc) to the figures and the legends to clarify. Please consider the images of E15.5 skeletal preps in Figure 1 and Figure 1—figure supplement 1 panels K-N. There is no ossification center in the humerus of the mutants, whereas all control and the *Tbx3fl/fl;Fgf8^mcm/mcm^*mutant (which have normal limbs) specimens have a well-defined ossification center.

In the text the authors conclude that this "reveals a role for Tbx3 in later aspects of Indian hedgehog regulated bone morphogenesis.” This has not been investigated to any extent and this type of comments goes far beyond what is actually shown in this paper.

We have removed this statement; it was a hypothesis that we did not test.

Also here the loss of the deltoid tuberosity need not have anything to do a direct effect on bone (cf Blitz, E, 2009).

We agree and did not make this statement.

Comparison of the phenotypes produced using the early acting (RARCre) and later acting (Prx1Cre) appears to show an effect of Tbx3 on the early stages of limb initiation that can be distinguished form later events of limb bud patterning through establishment of Shh in the posterior. A regulatory relationship between Tbx3, Hand2 and Shh has been reported in chick (Rallis, 2005).

We apologize for failing to cite this reference and have included it in the revision.

*Decreased expression of markers detected by RNAseq analysis is interpreted as evidence that fewer cells are undergoing condensation and chondrogenesis in posterior mesenchyme. This would be much better analyzing in a more direct way rather than making inferences from RNA-seq data.*

The lack of digit 5 chondrogenesis is shown in Figure 1 panel N and we have now confirmed the RNA-Seq data of decreased expression of Shh activated targets with qPCR (Figure 3 and Figure 3—figure supplement 4.

*Similarly, 5th digits agenesis is attributed to deceased cell number as opposed to decreased proliferation of digit 5 progenitors. These conclusions are drawn from a broad analysis of effects throughout the limb bud rather than focusing on the cells in question themselves.*

We have now specifically quantified proliferation in the posterior region containing d5 progenitors and added this data to Figure 3.

*Reviewer #3: The study described the limb phenotype in several Tbx3 conditional and null mutants in careful detail. Most of the conclusions based on whole mount RNA in situs are convincing and new findings. The most novel and surprising finding of the study is that the transcription factor TBX3 is localized to the cilium and interacts with KIF7. HOX proteins have been shown to interact with and inhibit GLI3 in the limb, however cilium localization was not explored. Given that the vast majority of TBX3 protein is in the nucleus (Figure 6), it seems surprising to me that the tiny amount in the cilium can be detected with IP (Figure 7). The interaction with GLI3 could be in the nucleus and/or cilium, but KIF7 should not be in the nucleus.*

We and others have shown (using different antibodies) that Kif7 is present in both the cilia and in scattered punctate regions of unknown composition in the cytoplasm and nucleus (Figure 4; He et al., 2014) We now also provide a confocal 100X maximum image projection showing this in sectioned limb bud and the z-stack (Figure 4—figure supplement 1 and Video 2).

The relative amounts of Kif7 in these different compartments have not been determined; most immunofluorescence images (including ours in Figure 4) highlight the signal in cilia. We have previously demonstrated (and do so again here) that Tbx3 is present in the cytoplasm and the nucleus, and that the relative abundance in these compartments is cell/tissue specific.

We do not claim that the IP’d material only reflects an interaction in the cilia and we think it likely that the relevant interactions between Tbx3, Gli3, Kif7 and Sufu are occurring at the ciliary base or centrosome as has been proposed by Ryan and Chiang and others (reviewed in Ryan and Chiang 2012; Ingham and McMahon 2009; and Wang et al., 2013; Wen et al., 2010, Humke et al., 2005; Tukachinsky et al., 2010). As discussed above, dissecting the cellular compartments in which different functionally relevant interactions has been a major, unsolved challenge in the field. What is clear is that processing of Gli3 to Gli3R requires the cilia. Our schematic specifically avoids indicating any localization for the Kif7/Tbx3/Gli3 complex (beyond anterior mesenchyme) because at this time, we can only hypothesize which subcellular compartment is functionally important. Determining this is clearly beyond the scope of this manuscript.

*To further bolster the most significant claim of the paper, I think that additional experiments are needed. In particular, the specificity of the antibodies must be proven (e.g. using Tbx2 mutant MEFs for the Tbx3 antibody) and ideally some results confirmed using tagged proteins. Tagged GLI3 is available and other proteins could be easily generated.*

We have performed these experiments, provide the data in the revised manuscript, and detailed the changes in response to Reviewers 1 and 2 above.

*A genetic test would be to determine whether heterozygosity for Kif7 in mice (or knockdown in cells) rescues the defect proposed to be due to increased KIF7.*

There is not an increase in the level of Kif7; we detected a change in localization in mutants.

*In addition, many of the claims of changes in gene expression based on the RNA-seq data in Tables 2 and 3 do not seem to be significant differences, and some are opposite to what is said in the Results. Thus, the value of the RNA-seq data, is not clear, especially as only two samples were analyzed for each genotype, although I appreciate it is a difficult experiment. Many of the additional points listed below are simple changes that would make the paper more accessible and data more convincing.*

We apologize for the inconsistencies between the text and the RNA-Seq tables; please see detailed explanations below.

*Figure 3': The mutant limb looks smaller than the control limb in (A), therefore the domain is expected to be smaller. The text implies the domain is smaller relative to the size of the limb. Which is the case?*

The text states “*Hand2* transcripts are reduced in E9.5 and E10.5 *Tbx3;PrxCre* mutantforelimb buds (Figure 2—figure supplement 1; Figure 3’)”. Based on the in situ images, the domain is both smaller and the intensity of the signal throughout the domain is decreased, which is consistent with fewer transcripts. We now provide qPCR data that support this claim (Figure 3). The mutant limbs are not in general smaller at E10.5-11.5; as noted above, the morphology and size varies, as it does among somite-matched wild type embryos, even within the same litter.

*"[…]downregulation of posterior genes including Shh, Shh pathway members, Tbx2, Sall1, Dkk1, and Osr1[…]": The decrease is very little (e.g. Sall1 -0.3).*

To address issues on the RNA-Seq data in general. Comments by both reviewers 2 and 3 led us to go back to the *eLife* website where we found that Tables 2 and 3 were duplicated and incorrectly labeled; the Table we referred to in the text was Table 2 when discussing the *posterior* expression analysis, but the table provided to reviewers labeled as Table 2 contained the *anterior* analysis. We sincerely apologize that the reviewers did not have access to files that were correct, and instead had files that were mislabeled relative to how they were discussed/referred to in the text, and that we failed to catch the error before the manuscript was sent for review.

Since initial submission, we have not had the time to generate sufficient animals to repeat the RNA-Seq experiment, but we have done additional validation of the datasets by qPCR (Figure 3, Figure 3—figure supplement 4). We have now labeled the tables with clear titles, provided methods that precisely describe how the changes contained in the tables were defined. Within the tables, we have labeled the data columns explicitly as mutant or control, and the compartments for the FPKM reads, as well as the pooled sample raw values (2 pools per genotype). We have also included additional data from an early microarray we performed in 2010 which in addition to the phenotypes, initially launched our investigation of the Shh pathway in the posterior and Gli3R in the anterior. This microarray analysis is in the present Table 2; changes that were replicated by the RNA-Seq experiment are highlighted in yellow in this table.

For the comparison of control posterior and mutant posterior (current Table 4), the fold changes in column L are given in log base 2, thus:

*Tbx2* -0.61, (decreased greater than 1.5 fold, statistically significant; also detected by microarray and confirmed by qPCR)

*Dkk1* -0.77, (decreased greater than 1.5 fold, statistically significant; also detected by microarray and confirmed by qPCR)

*Osr1* -0.89, (decreased greater than 1.5 fold, statistically significant; also detected by microarray and confirmed by qPCR)

*Cntfr* -0.93, (decreased greater than 1.5 fold, statistically significant; also detected by microarray and confirmed by qPCR)

On the current Table 4, *Sall1* is not listed because it was not changed more than +/-1.3 fold which was the criteria for inclusion on the table.

*"[…]up regulation of (AFP, Ttr, Apob, Apoa1, Apoa4, Trf, Ttpa)[…] (Table 2).”: Apob and Ttpa are actually down and the others are barely up in the anterior (Table 2).*

Please note, the manuscript paragraph was describing changes in the *posterior* mesenchyme and Table 2 should have been also *posterior*; as noted above, the reviewer was unfortunately evaluating the anterior analysis and thus it is understandable that the reviewer did not agree with our claims.

In the posterior dataset contained in the present Table 4, log base 2, column L:

Apob +2.7 (increased greater than 4 fold and statistically significant)

AFP +2.1 (increased greater than 4 fold and statistically significant)

Apoa4 +1.7 (increased greater than 2.5 fold and statistically significant)

Ttr +1.4 (increased greater than 2 fold and statistically significant)

Apoa1 +1.3 (increased greater than 2 fold and statistically significant)

Trf +0.6 (increased greater than 1.5 fold and statistically significant)

Ttpa +0.5 (increased greater than 1.5 fold and statistically significant)

However, based on the reviewers’ comments and the fact that we do not pursue these changes further, we chose not to make any claims about this set of transcripts in the revised manuscript.

More important, why are Grem1, Ptch1 and Gli1 not down in the RNA-seq, which would confirm the data in Figure 3 and address the question of whether the decrease is just due to the limb being smaller?

The mutant limbs are not in general smaller at E10.5-11.5; as noted above, the morphology and size varies, as it does among somite-matched wild type embryos in the same litter.

We performed thein situs showing decreased expression of these transcripts long before performing the RNA- seq experiment (the entire series of experiments took over 7 years to complete) because the phenotype and wealth of published data led us to consider alterations in Shh pathway activity as at least contributory to the loss of digit 5, and because the 2010 microarray detected the decrements in *Shh, Hand2* and *Gli1* transcripts. We cannot explain the failure of the RNA-Seq experiment to detect these differences. As this and the other reviewers noted, RNA-Seq is a screening test and must be validated for specific genes; in the case of *Shh, Grem1, Ptch1, Ptch2, Gli1, Hand2,* and many others, the in situ and qPCR data are consistent and clear.

Despite the failure of the RNA-Seq to detect some bona fide changes, we do not think that these RNA-Seq data have no value to us or to the field: aside from confirming and expanding on previously published anterior/posterior polarity datasets in the wild type limb (which we have now included in a separate Table 3: Differentially expressed transcripts detected by RNA-Seq of E11 control anterior forelimb versus control posterior forelimb buds), many expression changes detected in the control versus mutant datasets have been validated by multiple methods including in situ hybridization, microarray and qPCR and were also validated for effects on alternative splicing in our 2014 Plos Genetics paper (Kumar et al., 2014b, *eLife*).

*Results: "more single puncta cilia in mutants than controls (Figure 5, 84% vs 71%, p<0.001). The levels of Kif7 mRNA[…] unaffected[…] Tables 2, 3 […]": In Table 3 Kif7 is -0.3, which is greater than many of the changes in expression level claimed elsewhere in the text.*

See notes on the mix-up of the tables above; the original Table 3 reviewed contained the posterior gene expression data. In the current *anterior* analysis (Table 5), Kif7 is not listed on this Table because Kif7 log base 2-fold change is only 0.1042 (1.07 fold, not statistically significant) and only transcripts with changes +/- 1.3 fold (0.38 log base 2) are included on the revised tables.

*Results: "The presence of Tbx3 in cilia and response to Hedgehog pathway activation was confirmed with a commercially available anti-Tbx3 antibody and both SAG and Shh stimulation (Figure 6).": Tbx mutant fibroblasts should be shown to demonstrate specificity of the antibody.*

We agree and have responded as noted in response to other reviewers above. The specificity of our custom antibody has previously been published Frank et al., 2012, Frank et al., 2013 and is also shown in Figure 1 panel E/F and now, specifically relevant to ciliary localization in revised Figure 5 and Figure 6.

*Tagged proteins should be used to confirm the interactions thought to be detected with commercial antibodies.*

These data are now provided for the interactions between Tbx3 and both Kif7 and Gli3. The interactions we detected between Kif7 and Sufu and Gli3 proteins are well established in the literature.

*Figure 3: quantification is needed.*

We have now replicated these experiments and quantitated the changes in interactions using densitometry and the appropriate input controls. These data are in Figure 8 and its supplements.

*Results: "Consistent with our hypothesis, there was decreased interaction between Gli3FL and Sufu in mutants (Figure 6, lane 8[…]": Do the authors mean Figure 7? Also, since Gli3 FL and R are reduced, does this experiment say anything?*

We agree that the overall decrease in amount of Gli3 proteins in *Tbx3^△fl/^^△fl^*mutants makes it difficult to detect any decrement in protein/ protein interactions. We have not only assayed the Sufu/Gli3 interaction in both directions, but quantitated the amount of interaction and compared it to the decrease in protein levels (all in current Figure 8 and its supplements). In general, control embryos have ~ 1.6 fold more Gli3R than *Tbx3^△fl/^^△fl^* mutants (Figure 8 lanes 1-4, and Figure 8—figure supplement 2). However, the ratio of Gli3R interacting with Sufu in controls ranges from 4.3 to 9.5 fold higher than in mutants (Figure 8—figure supplement 2’). The interaction with Gli3FL is harder to assess reliably/quantitatively because of variable transfer efficiency due to high molecular weight. In the experiment shown in Figure 8, the ratio of Gli3FL in controls to mutants is 1.6-1.7, while the ratio of that IPd with Sufu is 4.6 fold greater in controls. When assaying for Sufu that coIPs with Gli3, the change is not as dramatic because it reflects the Sufu that IPs with all GLi3 species (Figure 8).

*Discussion: "Future studies will determine if Tbx3 transcriptional or posttranscriptional activity regulates production of a factor that represses expression or stability of Hand2 mRNA.": In the posterior limb, wouldn't Tbx3 also interfere with processing of Gli2/3 into activators, like in cilia mutants, and this could explain the posterior limb phenotype by the same mechanism as the anterior limb bud?*

As discussed in detail above, Shh signaling in the posterior limb bud is not cilia dependent; no ciliary mutants described thus far have posterior limb defects. Cilia mutants fail to process Gli3FL into Gli3R but as discussed above and in the manuscript, cilia phenotypes affect anterior digit number. Our data do not reveal any decrease in GliFL in the posterior, Shh responsive compartment.

[Editors’ note: the author responses to the re-review follow.]

*The manuscript has been improved but there are some remaining issues that need to be addressed before acceptance, as outlined below: 1) There are some remaining concerns about the quality of the biochemical data. Specifically, the composition and the order of loading of the lanes is not the same in Figure 4 and Figure 7, creating the impression that the shown immunoprecipitations are not from the same experiment. Gli3 and KIF7 blots should be included in Figure 7.*

We responded with this query on Jan. 26:

“We agree that it is not ideal that the order of the lanes is not consistent; nonetheless, the IPs were from the same experiment/limb bud lysates. These experiments were repeated at the request of the reviewer from the first submission. Since these IPs require literally hundreds of limb buds, and the results shown in the revision reproduce those originally presented, repeating them for the sake of changing the order of the lanes seems excessively burdensome and time/embryo consuming given that the conclusion is unchanged.”

We received this response from *eLife*:

*1) Figure 4 and Figure 7: Repeating the experiment just to present a consistent order of the lanes is indeed not necessary in this case since the material is difficult to obtain. The authors should state in the main text of the paper (if this is correct) that the lanes shown in these gels are from the same immunoprecipitations.*

We apologize that the order of the lanes is not consistent; the IPs were from the same experiment and that is now stated in the text.

Control blots for Gli3 and KIf7 from the same experiment should be included in Figure 7.

Now provided.

*Most importantly, error bars should be included in the plots showing the quantification of immunoprecipitation results in Figure 8 in order to illustrate that the experiment was performed more than once and the observed differences are significant.*

We sent this query on Jan. 26:

“To clarify, Figure 8 shows analysis of each protein-protein interaction performed once in each direction. The quantitation in the accompanying bar graph is for the blot shown and that is why the bar graphs do not have error bars. The repeat experiments showing that the altered interactions are reproducible, and their quantitations, are in Figure 8—figure supplement 2.

When considering these blots, it is important to realize that the IP and transfer efficiency do unavoidably vary from experiment to experiment, making it impossible to combine raw densitometry intensity data from all experiments to obtain the type of statistical analysis the reviewer is requesting. Furthermore, we think that the key point is that the changes in interactions in mutants are reproducible from experiment to experiment and when assayed in both directions.

As the reviewers are aware, it took 7 months to obtain sufficient numbers of embryos to provide the repeat experiments requested in the first review. We appreciate your reconsideration and additional clarification as to what it required for acceptance of our final submission.”

We received this response from *eLife*:

*“Quantification of immunoprecipitation results in Figure 8: It is obvious that to draw any conclusions about protein quantities on Western blots, each experiment should be performed at least twice. It is also obvious that the raw densitometry data cannot be compared between different experiments. However, band intensity ratios, which are currently presented per experiment, should be easy to combine and average between two or more experiments/samples, generating error bars. The authors should also label these plots appropriately (for example, instead of "ratio wt to mutant, lanes 1:2" use a label "Gli3FL, anti-Gli3-IP, ratio wt to mutant", etc.). Basically, instead of leaving the job of comparing different experiments to the reader, the authors should combine the quantitative outcome of their repeated experiments in one plot and include it in the main figures of the paper* ".

We have now remade Figure 8 and supplements in accordance with these requests. We have separated Figure 8 into sections entitled: Gli3/Sufu Interactions; Gli3/Kif7 Interactions; Kif7/Sufu Interactions

Within each of these sections, the upper blots are representative IP experiments that were repeated 3 times. Adjacent to these blots are bar graphs with the averaged band intensities and standard error of the mean. The lower blots in each section are the complementary IP/western performed in the “other direction”; these complementary experiments were done once, so the bar graphs only show the ratio of band intensities between control and mutant for that experiment.

So, for each interaction, we not only performed replicate experiments (some of which are presented in Figure 8—figure supplement 2), but also the complementary experiments. We hope the reviewers agree that this is a rigorous analysis for a methodology that is difficult to quantitate.

“*Please note that ratio of two bands cannot be negative (because if one positive value is divided by another positive value, the outcome is positive), so the values showing an increase in the mutant should be presented differently”.*

Thank you for the correction. We have remedied this in Figure 8 and Figure 8—figure supplement 2.

*2) Better discussion should be included to explain why the authors insist that TBX3 does not regulate GLI3 function in the posterior limb, and why they have ignored the highly related protein GLI2. If their hypothesis is correct, one would expect the same might be true for GLI2 since it also binds SuFu and requires cilia for proper processing. Also, if whole embryos reveal the same protein interactions as the anterior limb, why would the posterior limb be different? Gli2/3 double limb mutants could well have a worse posterior limb phenotype than Gli3 null or het mutants. Only a relatively late double conditional mutant of Gli2/3 has been published, and it indicated the posterior phenotype is worse than in single Gli3 mutants. Thus, some of the posterior phenotype in Tbx3 mutants could be due to the altered GLI processing and activity in this tissue, rather than just the decreased Shh expression, especially as Shh heterozygous mutants have normal limbs.*

We agree, and have considered and investigated this further. While *ShH^+^*/- mutants do not have an abnormal digit phenotype, and they have half as much *Shh* mRNA in their limb buds, the amount of protein is normal (Brian Harfe, personal communication/manuscript in revision) revealing a compensatory mechanism at the translational or post-translational level. Sanz-Ezquerro and Tickle (2000) proposed a compensatory, buffering system to stabilize polarizing activity in chick limb stating: “A buffering system can account for several regulative features of polarising region signalling. It can explain why limbs with normal patterns develop after application of extra Shh polarising region cells (Tickle et al., 1975), Shh expressing cells (Riddle et al., 1993) or Shh beads (Yang et al.,1997) to the posterior margin of chick buds, and why normal patterned limbs also develop after most, but not all, of the polarising region is removed (Fallon and Crosby, 1975; Pagan et al., 1996).”

Additionally, the *ShH^+^*/- genotype is sensitized to other mutations. For example, Zeller’s lab showed that rescued digits in Grem1-/-;Bmp4+/- are more sensitive to reduced Shh level (Benazet et al., 2009) and Hui's lab showed that Irx3/5 KO (excess HH activity) is improved by Shh dosage reduction. In addition, the Wnt7a mutant (Parr and McMahon, 1995), which affects Shh activation, has loss of digit 5. Mackem’s group re-evaluated the Wnt7a mutant (Zhu and Mackem, 2009), and found delayed onset and decreased level of *Shh* expression – which is consistent with our results.

In consideration of this information and to address the reviewers’suggestions, we have modified the model presented in Figure 9 and added the following paragraphs to the relevant sections of the Discussion.

For the section discussing posterior phenotype and mechanisms:

“Digit 5 formation is exquisitely sensitive to Shh activity and Grem1 (Harfe et al., 2004; Scherz et al., 2007; Zhu and Mackem, 2011; Zhu et al., 2008), so the altered expression of these genes in posterior mutant mesenchyme helps explain loss of this digit. […] Nonetheless, it is possible that decreased Shh signaling in the posterior of *Tbx3;PrxCre* mutant limb buds creates changes in Gli3 levels/ratios below our ability to detect. If so, decreased Shh activity would be predicted to increase Gli3R, decreasing digit number.”

For the section discussing anterior phenotype and mechanisms we have added:

“Loss of Tbx3 could also influence stability and processing of Gli2. The anterior PPD phenotype of *Gli3* heterozygotes is slightly more severe in a *Gli2* null background (Bowers et al., 2012; Mo et al., 1997). […] This is consistent with our findings that Tbx3 affects anterior digit number by regulating Gli3 repressor stability in the anterior mesenchyme.”

Finally, it seems to be an over-interpretation of the data to say that the posterior limb is not dependent on cilia. In cilia mutants’ production of both activators and repressors is altered (diminished), but obvious phenotypes are only seen where the balance is greatly skewed. It would be nice to extend the Discussion of your paper to address these points.

We agree; in the manuscript, we discuss what is known with regard to ciliary function and limb phenotypes but do not exclude the possibility of posterior functions yet to be determined.

[Editors' note: further revisions were requested prior to acceptance, as described below.]

*The manuscript has been improved but there are some remaining issues that need to be addressed before acceptance, as outlined below: 1) The scan of the Western blot shown in Figure 7 is not publication quality. Please substitute it for a high-resolution scan.*

We have replaced the image in 7C with that exported at 600 dπ from the scanner; we have shown the original image without any alterations to lane order or intensity. Because of the magnification at which the image was originally taken, the Biorad scanner gives a pixelated quality to the over exposed IgG bands and unfortunately, this is the lowest exposure we have of this experiment. We have provided additional experiments showing the Gli3/Tbx3 interaction in the new Figure 7—figure supplement 1.

*Also, please make sure that in all cases where lanes of Western blots have been left out, a clear separator line is present (this currently seems not to be the case in Figure 7 and Figure 8).*

For Figure 7, the new image shown was exported at 600 dπ from the scanner; we have shown the original image without any alterations to lane order or intensity, thus there are no lane rearrangements requiring separator lines. We have added more obvious separators to Figure 7and Figure 7 and presented these blots with lane order consistent with that of Panel 7C. We have added the requested separator to Figure 8.

2. Please reconsider the title of your paper. "T-box3 is a ciliary protein and regulates.." doesn't read well. Furthermore, eLife encourages the authors to provide brief explanations for the acronyms used in the title and Abstract. For your paper, mentioning that Gli3 is a player in Sonic Hedgehog signaling would be appropriate.

We have revised the title to: “T-box 3 is a Ciliary Protein and Regulates Stability of the Gli3 transcription factor to Control Digit Number”. We think that including Sonic Hedgehog signaling in the title is too cumbersome and also, inaccurate because its function in the anterior is Shh independent, as stated in the Abstract. We are open to suggestions if the title is still not satisfactory.

We have revised the Abstract as requested to include additional reference to Gli3 as a transcriptional effector in the Hedgehog pathway:

“Crucial roles for T-box3 in development are evident by severe limb malformations and other birth defects caused by T-box3 mutations in humans. […] Remarkably, T-box3 is present in primary cilia where it colocalizes with Gli3. T-box3 interacts with Kif7 and is required for normal stoichiometry and function of a Kif7/Sufu complex that regulates Gli3 stability and processing. Thus T-box3 controls digit number upstream of Shh-dependent (posterior mesenchyme) and Shh-independent, cilium-based (anterior mesenchyme) Hedgehog pathway function.”